# Chain-of-Model Learning for Language Model

**Xiaohua Wang**[1,2,*]    **Kaitao Song**[2,*,†]    **Xu Tan**[*]    **Huiqiang Jiang**[2]    **Chengruidong Zhang**[2]
**Yongliang Shen**[3]    **Cen Lu**[2,5,6]    **Zihao Li**[2,7]    **Zifan Song**[2,8]
**Caihua Shan**[2]    **Yansen Wang**[2]    **Kan Ren**[2,4]    **Xiaoqing Zheng**[1]
**Tao Qin**[2]    **Yuqing Yang**[2]    **Dongsheng Li**[2]    **Lili Qiu**[2]

Fudan University[1]    Microsoft Research[2]    Zhejiang University[3]    ShanghaiTech University[4]
Idiap Research Institute[5]    EPFL[6]    UIUC[7]    Tongji University[8]

## Abstract

In this paper, we propose a novel learning paradigm, termed *"Chain-of-Model"* (CoM), which incorporates the causal relationship into the hidden states of each layer as a chain style, thereby introducing great scaling efficiency in model training and inference flexibility in deployment. We introduce the concept of *"Chain-of-Representation"* (CoR), which formulates the hidden states at each layer as a combination of multiple sub-representations (i.e., chains) at the hidden dimension level. In each layer, each chain from the output representations can only view all of its preceding chains in the input representations. Consequently, the model built upon CoM framework can progressively scale up the model size by increasing the chains based on the previous models (i.e., chains), and offer multiple sub-models at varying sizes for elastic inference by using different chain numbers. Based on this principle, we devise *Chain-of-Language-Model* (CoLM), which incorporates the idea of CoM into each layer of Transformer architecture. Based on CoLM, we further introduce CoLM-Air by introducing a *KV sharing* mechanism, that computes all keys and values within the first chain and then shares across all chains. This design demonstrates additional extensibility, such as enabling seamless LM switching, prefilling acceleration and so on. Experimental results demonstrate our CoLM family can achieve comparable performance to the standard Transformer, while simultaneously enabling greater flexiblity, such as progressive scaling to improve training efficiency and offer multiple varying model sizes for elastic inference, paving a a new way toward building language models. Our code will be released in the future at: `https://github.com/microsoft/CoLM`.

## 1    Introduction

With the advent of large language models (LLMs) [1–6], scaling up Transformer architecture [7–9] has been considered as the de facto approach to revolutionize existing AI landscape and achieve state-of-the-art performance across a wide range of different tasks. Therefore, there has a growing trend in both industry and academia to explore strategies for scaling up Transformer models. Under these circumstances, the parameter size of LLMs has been an exponential growth, scaling from the billion to the trillion level. As a result, its explosive parameters also impose extremely expensive burden on training and cannot offer varying inference usage for different deployment environments. In light of this increasing trend of scaling laws, how to develop and harness LLMs effectively to address user instructions in various scenarios has been an open and critical challenge among the whole community.

---

[*] The first three authors have equal contributions.
[†] Corresponding author: Kaitao Song (*stillkeeptry@outlook.com*)

39th Conference on Neural Information Processing Systems (NeurIPS 2025).

Motivated by elastic inference [10, 11] and continual training [12, 13], we claim that existing scaling strategies for LLM architectures still remain some inherent issues:

- Unlike human intelligence, which acquires new knowledge incrementally, existing scaling policies cannot retain existing scales and always requires training from scratch, leading to inefficiencies.
- Existing LLM architectures (e.g., Dense or MoE [14]) always activate a fixed scale of parameters, lacking mechanisms to dynamically adapt problem-solving capabilities.

In this paper, we propose a new concept, termed "*Chain-of-Representation*" (CoR), that generalizes the scope of representation paradigm to a broader range. Specifically, we observe that any representation can always be viewed as a combination of multiple sub-representations at the hidden dimension level. Therefore, we define this composition as Chain-of-Representation, that each sub-representation corresponds to one chain. Based on this definition, by using different number of preceding chains, its corresponding features can be used to encode different knowledge (we call it "scale"), just as shown in Figure 1. Therefore, how to build the connections among CoR features to ensure feature transformation across each scale has been a critical step.

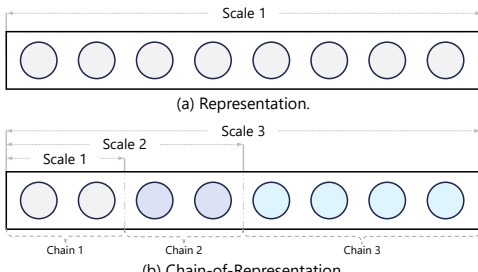

Figure 1: An example of CoR with 3 chains.

To this end, we introduce "*Chain-of-Model*" (CoM), a novel learning paradigm for modeling CoR features. Its core idea is to incorporate causal dependencies across different scales, ensuring that each scale can only use information from its preceding scales at the representation level. To fulfill this, we propose "*Chain-of-Layer*" (CoL), to reformulate current network layers based on CoR features. Specifically, we require each output chain $i$ is only conditioned on the input chains from 1 to $i$. Figure 2 illustrates an example of applying CoL into Linear layer. CoL is capable of three important characteristics: *Generality*, *Causality* and *Compositionality*. That means, 1) Any layer can be viewed as CoL (chain is 1); 2) To obtain the features at scale $i$, CoL allows us to only activate the parameters of its correlated chains; 3) If two layers both conform to the CoL settings, their composition also adheres to the CoL property. Following this principle, we can stack multiple CoL layers to generalize it from the layer level to the model level, i.e., Chain-of-Model (CoM). The scope of CoM can be generalized to any ML models as all models can be considered as a special case of CoM that chain is set as 1. Besides, the output features of CoM also conform to the characteristic of CoR, CoM can offer multiple sub-models at different scales by using different number of chains. Based on these characteristics, CoM can exhibit greater flexibility to build foundation models.

On the basis of CoM framework, we re-formulate the architecture of language models by applying the idea of CoL into each layer within Transformer, and term it as "*Chain-of-Language-Model*" (CoLM). CoLM can integrate multi-scale training within single forward propagation, to offer multiple sub-models for elastic inference. It also enables us to scale up language models via chain expansion (i.e., increase the number of chains), while simultaneously preserving the capability to use previous scales. Moreover, based on the CoL criteria, we further introduce a *KV sharing* mechanism in attention module that requires all keys and values to be computed within the first chain, and term it as CoLM-Air. Based on this mechanism, CoLM-Air offers greater extensibility and flexibility, including allowing us to seamless switching any different scales of LLMs without any additional re-computations (e.g., keys and values), accelerating prefilling within the first chain and so on. Experimental results on multiple benchmarks also demonstrate our CoLM family can achieve comparable performance while both exhibit better extensibility and flexibility. Overall, the contributions of our paper can be summarized as below:

- We point out the inherent issues of existing LLM scaling approaches, and formulate the concept of chain-of-representation (CoR), that integrates multi-scale capabilities within single representation.
- To model the relationships among CoR features, we introduce chain-of-model (CoM) learning and further extend it into language model, termed CoLM. CoLM is able to provide multiple sub-models at varying scales within a unified foundation model, enabling more flexible and scalable extensions.
- Experimental results on multiple benchmarks demonstrate that our CoLM family can achieve comparable performance to existing LLM frameworks, while simultaneously exhibit better extension extensibility and flexibility in different scenarios (e.g., prefilling, tuning and so on).

## 2 Chain-of-Model Learning

In this section, we first introduce the concept of "*Chain-of-Representation*" (CoR), which generalizes the paradigms of existing representation to a broader scope. After that, we introduce "*Chain-of-Layer*" (CoL) and "*Chain-of-Model*" (CoM), to model CoR from the layer level to the model level.

### 2.1 Representation

> **Definition 1 (*Chain-of-Representation*)** *For representation $x \in \mathbb{R}^D$, it can always be equivalently formulated as a concatenation of $n$ sub-representations, denoted as $\xi(x, n) = \{x_1, \ldots, x_n\}$, where $x_i \in \mathbb{R}^{d_i}$ and $\sum_{i=1}^{n} d_i = D$. We term this as chain-of-representation (CoR) of $x$.*

According to Definition 1, each chain corresponds to each sub-representation (i.e., $x_i$) in CoR. By activating the first few chains (i.e., $x_{<i}$), it can be used to encode information of $scale_i$, with a dimension of $sum(d_{\leq i})$. Hence, CoR allows us to encode n different scales within one representation. If $n = 1$, CoR is identical to the original representation. Figure 1 illustrates the CoR.

### 2.2 Layer

However, a challenge arises is how to design layers to build connections between CoR inputs and CoR outputs, enabling multi-scale feature transformation while both preserving the output features to follow the criteria of CoR in Definition 1. Therefore, we need to maintain each scale can only utilize information from all its preceding scales, and introduce *Chain-of-Layer* to incorporate causal relationships into hidden states of CoR, as below:

> **Definition 2 (*Chain-of-Layer*)** *For a layer $y = f_\theta(x)$, where its input $x$ and output $y$ can both be expressed as the chain-of-representation $\xi(x, n)$ and $\xi(y, n)$. We term it as a Chain-of-Layer (CoL) where $f_\theta(\cdot)$ satisfies that each $y_i$ is only conditioned on $x_{\leq i}$.*

CoL is capable of three fundamental properties – *generality*, *causality* and *compositionality*:

> **Corollary 2.1 (*Generality*)** *Any layer can be viewed as a case of chain-of-layer when $n = 1$.*

If $n = 1$, CoL can be considered as the standard network layer to process the conventional representation. In other words, any network layer can be generalized to the form of CoL. Therefore, we can gradually increase layer dimension by introducing additional chains upon the existing chains.

> **Corollary 2.2 (*Causality*)** *If a layer $y = f(x)$ satisfies the property of chain-of-layer, that means its weights $\theta$ can be partitioned into $n$ independent parts $\{\theta_1, \ldots, \theta_n\}$, where each $\theta_i$ is used to compute $y_i$ based on $x_{\leq i}$. Therefore, $y$ allows the causal computation, i.e., to obtain feature $y_i$, we only need to compute $\theta_{\leq i}$ based on $x_{\leq i}$.*

In the CoL design, each scale can aggregate information of all its previous scales to increase scalability and effectively avoid catastrophic forgetting [15]. Therefore, to obtain information of $Scale_i$, this design allows us to only compute its preceding chains (i.e., $\theta_{\leq i}$), without calculating the whole weights. As a result, CoL can offer multi-scale information within one representation and allow us to dynamically calculate features at varying scales.

> **Corollary 2.3 (*Compositionality*)** *Assume we have two layers $y = f_1(x)$ and $z = f_2(y)$, where $x = \xi(x, n)$, $y = \xi(y, n)$ and $z = \xi(z, n)$. If both $f_1$ and $f_2$ adhere to the requirements of CoL, their composition $z = f_2(f_1(x))$ also satisfy the setting of CoL, i.e., each $z_i$ only depends on $x_{\leq i}$.*

More important, CoL also supports compositionality, meaning that stacking multiple CoL layers also retains the characteristics of CoL. This property allows us to generalize the scope of CoL from the layer level to the model level.

## 2.3 Model

> **Definition 3 (*Chain-of-Model*)** *For a model $\theta$ with $L$ layers, we define it as a **chain-of-model** (CoM) if each of its layers, from the first to the final, conforms to the chain-of-layer property.*

Based on Definition 3, if a model satisfies the criteria of CoM, it also inherits all CoL properties, such as generality and causality. In other words, any model can be viewed as a kind of CoM (i.e., $n = 1$). And CoM can integrate multiple sub-models at different scales within one model, and enable us to scale up model from previous capabilities. This capability directly empowered foundation models with better extensibility and flexibility. In the next section, we will investigate how to build language model based on CoM framework.

# 3 Architecture

In this section, we describe the details about how to apply CoM into language model, including Linear, each module (e.g., embedding, self-attention, feedforward, normalization) within Transformer, and the objective function. Here, we term it as Chain-of-Language-Model (i.e., CoLM). Moreover, we further introduce a KV sharing mechanism based on CoLM framework, and term it as CoLM-Air, which provides better flexibility, such as prefilling speedup and seamless LLM switching, which always needs to re-calculate keys and values..

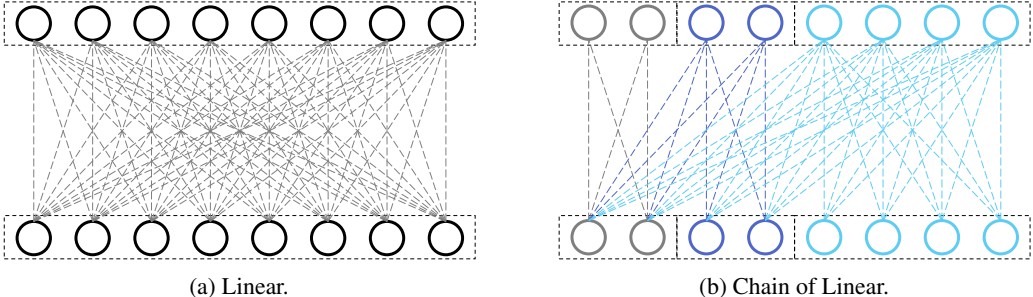

(a) Linear.          (b) Chain of Linear.

Figure 2: Comparisons between Linear and Chain-of-Linear layer.

## 3.1 Linear

Linear layer is a fundamental layer in neural network, which offers linear operation to transform the input features into the output features. The mathematic expression of linear layer is as $y = Wx + b$, where $W \in \mathbb{R}^{D_y \times D_x}$ and $b \in \mathbb{R}^{D_y}$. Here, each neuron of $y$ is dense connected to the input feature $x$, so that we always need to compute all elements of $W$ and $b$ to preserve the integral semantics of $y$. Following the CoL setting, we introduce a new hyper-parameter into the linear layer, termed "*Chain*", to determine the size of each chain in $x$ and $y$ as represented by CoR. Here, we set the hyper-parameter $\mathcal{C} = \{c_1, \ldots, c_n\}$ as the base ratio to compute the dimension of each $x_i \in \mathbb{R}^{D_{x_i}}$ and $y_i \in \mathbb{R}^{D_{y_i}}$, where $D_{x_i} = \frac{c_i}{\sum \mathcal{C}} \cdot D_x$ and $D_{y_i} = \frac{c_i}{\sum \mathcal{C}} \cdot D_y$. Therefore, it is calculated as:

$$y = \Big\|_{i=1}^{n} (y_i) = \Big\|_{i=1}^{n} (W_i x_{\leq i} + b_i), \tag{1}$$

where $W_i \in \mathbb{R}^{D_{y_i} \times (\sum_{j=1}^{i} D_{x_j})}$, $b_i \in \mathbb{R}^{D_{y_i}}$ and $\|$ denotes the concatenation operation. Given this design, the linear layer also satisfies the CoL criteria, and is identical to the standard linear when $n = 1$. Here, we term it as *Chain-of-Linear* or *Linear (Chain)* layer, which serves as the foundation to build language model. Figure 2 depicts the comparisons between Linear and Chain-of-Linear layer. To facilitate large-scale pre-training, we further provide an efficient implementation with a well-designed block-wise sparse kernel, optimizing both data access and all-reduce communication, as detailed in Appendix A.6.1.

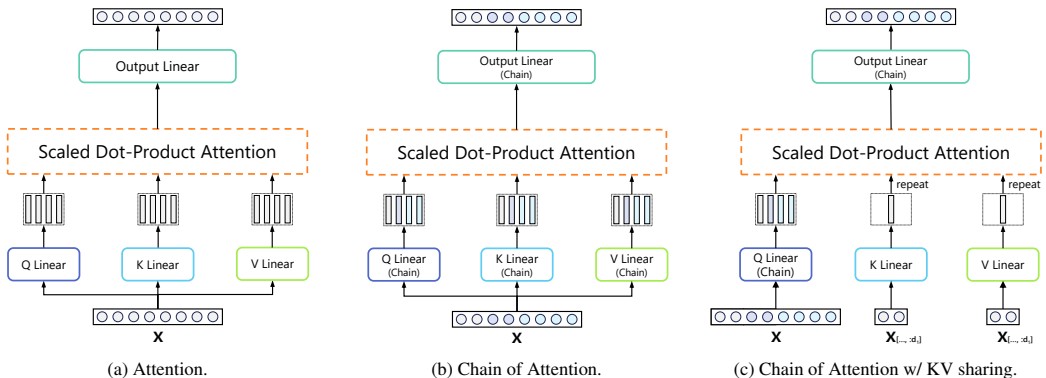

| (a) Attention. | (b) Chain of Attention. | (c) Chain of Attention w/ KV sharing. |

Figure 3: Differences between Attention, Chain of Attention and Chain of Attention with KV sharing.

## 3.2 Transformer

### 3.2.1 Multi-head Attention

In MHA, it applies linear operation to first obtain the query, key and value based on the input features, and split each into multiple heads. And then, we perform the attention function over each head (i.e., query, key and value) and concatenate all heads, followed by a linear transformation to produce the final output. Overall, the mathematic expression of MHA can be formulated as:

$$\mathrm{MHA}(x) = \mathrm{O}\big(\|_{i=1}^{h}\mathrm{Attention}(q_i, k_i, v_i)\big) = \mathrm{O}\big(\|_{i=1}^{h}\mathrm{softmax}(\frac{q_i k_i^{T}}{\sqrt{d_k}})v_i\big),$$
$$\text{where } \|_{i=1}^{h}q_i = Q(x), \|_{i=1}^{h}k_i = K(x), \|_{i=1}^{h}v_i = V(x), \tag{2}$$

where $h$ is the head number, $d_k$ is the head dimension, and $Q(\cdot), K(\cdot), V(\cdot), O(\cdot)$ represent Linear layer to obtain query, key, value and output. To support CoR modeling in the MHA, we first directly replace all Linear layers (i.e., Q, K, V, O) by using Chain-of-Linear layers described in Sec 3.1. However, due to the dot product operations within $\mathrm{Attention}(q, k, v)$, if a single head (i.e., $q_i$, $k_i$, $v_i$) contains information from 2 or more chains, its output will blend these information and then distribute to each chain, which cannot meet the setting of CoL. Therefore, we expect each head to only contain single scale information. To fulfill this, we design a simple trick that requires the sum of $\mathcal{C}$ is equal to $h$, so that each $c_i$ represents how many heads will be allocated for chain $i$. This design enables each chain to retain their dedicated query, key, and value, ensuring our attention can satisfy the criteria of CoL, and we term it as *Chain-of-Attention* or *Attention (Chain)* module. Figure 3 illustrates the differences between Attention and Chain-of-Attention.

### 3.2.2 Feed-Forward Network

Feed-Forward Network (FFN) is another crucial component of the Transformer, which is composed of multiple linear layer interleaved with non-linear activation functions (e.g., ReLU [16] or SiLU [17]). To generalize the CoM framework into Transformer, FFN should also comply with the setting of CoL. To achieve this, we just need to substitute each Linear of FFN by using Chain-of-Linear layer, and adopt the same hyper-parameter $\mathcal{C}$ used as Attention (Chain) module. In this way, the output features of FFN can also conform to the requirements of CoR. The implementation and illustrative example about FFN can refer to Appendix A.6.3.

### 3.2.3 Normalization

Normalization is a technique to stabilize layer-wise training. In Transformer, we perform normalization [18, 19] at the feature level, before attention or FFN layers. To preserve the CoR properties, we apply the normalization function to each chain individually. Specifically, we use Root Mean Square Layer Normalization (RMSNorm) [19] for each chain to align with the LLaMA architecture. This chain-wise operation not only guarantees that nor-

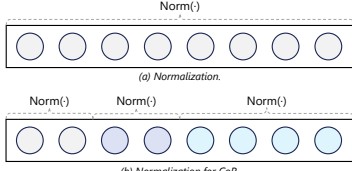

Figure 4: Normalization (CoR).

malized features maintain their causal structure but also slightly
improves training efficiency by enabling group-wise operations, which can reduce the number of
all-reduce communications. Figure 4 depicts the design of normalization layer.

### 3.2.4 Embedding

Embedding $\mathbf{E} \in \mathbb{R}^{V \times D}$ usually serves as the first layer in language model that converts the raw data
$\mathbf{I}$ as the input embeddings $x \in \mathbb{R}^D$. We do not involve any modifications at the Embedding during the
training, but when encoding information at scale $i$, we just need to use embeddings $\mathrm{x}[: \mathrm{sum}_{j=1}^i \mathrm{D_{x_j}}]$
corresponding to the first $i$ chains. A detailed example can be found in the Appendix A.6.4.

### 3.3 KV Sharing

Based on our above designs, Transformer can provide multiple capability across different models
scales, enabling flexible deployment for various applications. However, in attention module, each
chain also preserves its unique set of keys and values, and cannot bridge the connections between
different scales. For example, when transitioning from a small language model (SLM) to a LLM
for generation, it typically requires re-computing all keys and values for the preceding content. To
this end, we further propose a ambitious technique, called *KV sharing*, where all keys and values
are computed only in the first chain, and then shared to all other chains. If the number of keys and
values is smaller than query heads, we follow the practices of GQA [20] by repeating keys and values
to match the query heads, just as shown in Figure 3c. The design also meets the criteria of CoL [1].
Although it slightly affect the performance, this design also exhibits some unique characteristics,
e.g., prefilling speedup, and allowing us to seamlessly switch LMs at different scales from CoLM for
generation. To the best of our knowledge, this is the first work to fulfill this functionality, and we term
our CoLM with KV sharing as CoLM-Air. More detailed implementation are in Appendix A.6.2.

### 3.4 Objective Function

Based on the above designs of each component, the output feature $x \in \mathbb{R}^D$ of Transformer network
will adhere to the CoR form $\xi(x, n)$. And finally, we apply a classification layer $\mathbf{W} \in \mathbb{R}^{D \times V}$, to
map the feature dimension into the vocab size. Generally, we can adopt the cross entropy loss as the
objective function for CoLM training. However, to enable multi-scale prediction, we also need to
build individual classification heads for each scale. Motivated by MRL [21], we propose a multi-chain
cross-entropy loss to compute each scale as: $\mathrm{Loss_i} = \mathcal{L}(\mathrm{W^i x_{\leq i}})$, where $\mathrm{W^i} \in \mathbb{R}^{\mathrm{sum}_{j=1}^i (\mathrm{D_{x_j}}) \times V}$ is
a subset of $\mathrm{W}$, and $\mathcal{L}$ is the cross entropy function. Although the computations of different logits
(i.e., $\mathrm{W^i x_{\leq i}}$) can be shared without any additional cost, calculating multiple cross-entropy losses will
still bring some computational overhead during the training. Due to resource limitations, we adopt a
two-stage strategy inspired by MRL. We first pre-train the entire CoLM network using a standard
cross-entropy loss calculated only on the output of the final, largest scale. This allows the model
backbone to converge efficiently. After pre-training, we freeze the model backbone and efficiently
fine-tune the classification heads for all scales simultaneously.

## 4 Experiments

### 4.1 Performance

In our experiments, we utilize the SlimPajama dataset [22] as the pre-training corpus, comprising 600
billion tokens. Here, we use LLama-2-tokenizer [2] to process corpus, with a vocab size of 32000. We
leverage Fully Sharded Data Parallel (FSDP2) for distributed training, facilitated by the TorchTitan
framework [23]. Our training is built upon a cluster of 32 NVIDIA A100 40GB GPUs with a gradient
accumulation of 4, yielding an effective batch size of 1024. BFloat16 is adopted for training, and
the sequence length is set as 4096 tokens. We choose AdamW optimizer [24] with a learning rate of
$1.5 \times 10^{-4}$. Due to resource limitations, our model is pre-trained with 50K steps, nearly 200 billion
tokens. For model configuration, we use chains $\mathcal{C}$ and dimensions $D$ to determine different models.
The baseline use $\mathcal{C} = \{32\}$, and adopt same configuration as LLaMA-3.2-1B, so it is equivalent to

---

[1]It can be considered as $y_i$ is only conditioned on $x_1$, which belongs to a subset of Definition 2.

[2]`https://huggingface.co/meta-llama/Llama-2-7b`

the standard language model. For CoLM-series, we use $\mathcal{C} = \{16, 16\}$ and $\mathcal{C} = \{8, 8, 8, 8\}$ [3]. To align the baseline, we also increase the dimension of our CoLM as Chain-of-Linear takes fewer parameters than Linear layer. We use the EleutherAI Language Model Evaluation Harness [25] to evaluate models on commonsense tasks in a zero-shot setting. More details can refer to Appendix A.1.

Our results are reported in Table 1, and have these observations: 1) CoLM-Air with KV sharing mechanism will slightly affect model performance, as it only calculates keys and values within the first chain. However, this design introduces greater flexibility, which will be discussed in the following section; 2) using $\{16, 16\}$ can achieve better performance when compared with $\{8, 8, 8, 8\}$. When using $\{16, 16\}$ with a wider size, our method can achieve comparable performance to baseline. It can be considered as a trade-off as using more chains can offer more sub-models for usage. Overall, CoLM achieves results competitive with baseline while offering significantly faster prefilling speeds and greater flexibility, which will be discussed in subsequent sections.

Table 1: Performance of baseline and CoLM on commonsense reasoning tasks. Models are trained for 50K steps. Each task is reported by acc_norm metric.

| Chain($\mathcal{C}$) | Dims | Params | HellaSwag↑ | Obqa↑ | WinoGranda↑ | ARC-e↑ | ARC-c↑ | Boolq↑ | Piqa↑ | Avg↑ |
|---|---|---|---|---|---|---|---|---|---|---|
| {32} | 2048 | 1.10B | 40.01 | 31.19 | 52.72 | 43.52 | 23.63 | 57.43 | 67.30 | 45.11 |
| *CoLM* | | | | | | | | | | |
| {16, 16} | 2560 | 1.11B | 40.25 | 31.39 | 52.41 | 43.73 | 23.81 | 58.01 | 67.30 | 45.27 |
| {16, 16} | 2048 | 0.86B | 37.12 | 29.18 | 51.07 | 40.82 | 22.61 | 61.74 | 65.48 | 44.00 |
| {8, 8, 8, 8} | 3072 | 1.18B | 38.63 | 31.99 | 51.62 | 42.80 | 23.89 | 56.70 | 66.00 | 44.51 |
| {8, 8, 8, 8} | 2048 | 0.74B | 34.49 | 27.57 | 50.20 | 39.02 | 22.78 | 57.65 | 63.00 | 42.10 |
| *CoLM-Air* | | | | | | | | | | |
| {16, 16} | 2560 | 1.11B | 39.85 | 31.19 | 52.09 | 44.30 | 23.63 | 56.72 | 66.76 | 44.80 |
| {16, 16} | 2048 | 0.86B | 36.82 | 28.77 | 51.62 | 49.19 | 22.70 | 61.31 | 65.94 | 43.90 |
| {8, 8, 8, 8} | 3072 | 1.18B | 36.98 | 29.78 | 49.80 | 41.46 | 24.57 | 55.81 | 65.51 | 43.41 |
| {8, 8, 8, 8} | 2048 | 0.74B | 33.77 | 27.97 | 49.41 | 38.93 | 22.61 | 57.65 | 62.10 | 41.77 |

## 4.2 Chain Expansion

Table 2: Performance of baseline versus expanded models on commonsense reasoning tasks.

| Model | HellaSwag↑ | Obqa↑ | WinoGranda↑ | ARC-e↑ | ARC-c↑ | Boolq↑ | Piqa↑ | Sciq↑ | Avg↑ |
|---|---|---|---|---|---|---|---|---|---|
| *Tiny-LLaMA-v1.1* | | | | | | | | | |
| Baseline | 61.47 | 36.62 | 59.43 | 55.47 | 32.68 | 55.99 | 73.56 | 84.20 | 57.43 |
| + Expansion | 61.66 | 36.62 | 60.62 | 56.27 | 32.94 | 58.44 | 74.05 | 86.20 | **58.35** |
| *LLaMA-3.2-1B* | | | | | | | | | |
| Baseline | 63.78 | 36.42 | 59.83 | 60.56 | 36.03 | 63.70 | 74.37 | 88.40 | 60.39 |
| + Expansion | 63.19 | 37.02 | 60.69 | 60.31 | 36.69 | 63.55 | 74.65 | 88.20 | **60.53** |

Given the generality and causality of CoM, any model can be regarded as a special case of CoM when chain is 1, and can be extended to multiple chains. Therefore, we introduce *Chain Expansion*, which use the well-trained model as the initial chain and then expand it with new additional chains. Conceptually, this is similar to progressive network architectures [12], that allowing us to increase additional capacity while retaining prior knowledge. However, CoLM has extended it to Transformer for training. Overall, *Chain Expansion* allows us to incrementally increase model scales based on the existing scales, avoid training from scratch.

To validate this point, we choose two LLaMA variants (i.e., TinyLLaMA-v1.1 [26] and LLaMA-3.2-1B [3]) as the first chain for expansion. Since these two models both use group query attention [20], where query number is 32 and kv head is 8. Following the Attention setting in Section 3.2.1, We set $c_1 = 32$ and subsequently introduce a second chain with $c_2 = 8$, incurring nearly 0.8B parameters. We perform zero initialization for the expanded classification head, so that its output logits is started from the original trained model. Due to resource limitations, our expanded models are trained with a batch size of 1024 for 20K steps, nearly 8 billion tokens. Besides, we also freeze the first chain to preserve the original knowledge and speedup training. More details can refer to Appendix A.1.

All of results are presented in Table 2. We achieve improvements of $0.92$ and $0.14$ over the Tiny-LLaMA-v1.1 and LLaMA-3.2-1B, respectively. Since LLaMa-3.2-1B is a stronger baseline, it

---

[3]Each chain is equal can slightly improve the training efficiency.

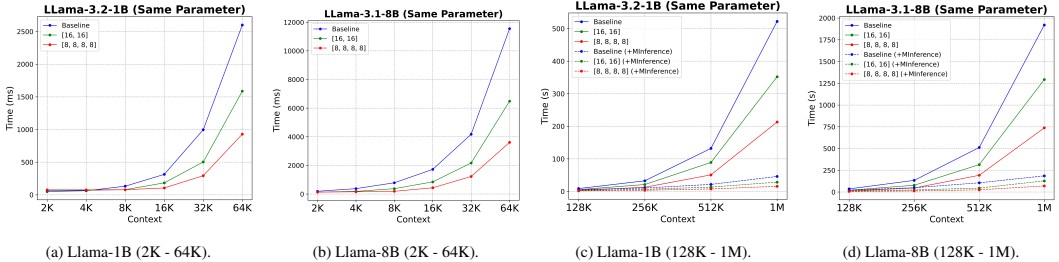

|  (a) Llama-1B (2K - 64K). | (b) Llama-8B (2K - 64K). | (c) Llama-1B (128K - 1M). | (d) Llama-8B (128K - 1M). |

Figure 5: Comparisons on Prefilling speed on LLaMa-1B and LLaMa-8B settings (2K - 1M).

requires more computation to achieve significant gains but our method still can improve it under the limited computations. Overall, these results also indicate the effectiveness of our method in improving the baselines, even with constrained resources.

## 4.3 Elastic Inference

Elastic inference is to provide dynamic inference usage to meet the requirements from different deployment scenarios. Compared with previous methods [10, 11] that involved additional sampling policies, CoLM integrates multiple sub-models at varying scales within one model, enabling unified pre-training via a single forward pass. To obtain sub-models across different scales for prediction, we further pre-train our CoLM model using the loss function described in Sec 3.4. We choose the setting of CoLM-Air, where $\mathcal{C} = \{16, 16\}$ and Dims as 2048, for validation. Following prior work [21], we freeze the Transformer backbone and only train the classification head to accelerate the training. The results in Table 3 also highlight the potential of CoLM for enabling elastic inference.

Table 3: Elastic Inference to offer multiple sub-models by using different number of chains.

| Scale | Params | HellaSwag | Obqa | WinoGranda | ARC-e | ARC-c | Boolq | Piqa | Avg |
|---|---|---|---|---|---|---|---|---|---|
| CoLM-Air, $\mathcal{C} = \{16, 16\}$, $\mathrm{Dims} = 2048$ | | | | | | | | | |
| Chain 1 + 2 | 0.86B | 36.82 | 28.77 | 51.62 | 40.19 | 22.70 | 61.31 | 65.94 | 43.90 |
| Chain 1 | 0.33B | 29.31 | 25.75 | 52.96 | 34.50 | 22.01 | 62.23 | 61.15 | 41.13 |

## 4.4 Prefilling

Prefilling [27] is to generate keys and values in Transformer to understand the prompt of user instruction. Existing LLM paradigms always requires us to deploy the full model for calculation, introducing substantial computational costs. For CoLM-Air, all keys and values are computed within the first chain, so that we just need to calculate the first chain according to the causality of CoM. Please note the first chain is still a standard language model, we can further apply any inference-time techniques (e.g, MInference [28]). Here, we conduct two scales (LLama-3.2-1B and LLama-3.1-8B) for evaluation, with a context length from 2K to 1M. We choose two settings where $\mathcal{C} = \{8, 8, 8, 8\}$ and $\mathcal{C} = \{16, 16\}$ with similar parameters as baseline. Our results are shown in Figure 5. More experimental details and results can refer to Appendix A.2.1.

From Figure 5, We observe that CoLM-Air achieves faster prefilling speed when compared to LLaMa with similar parameter sizes. With the increasing of sequence length, CoLM-Air can achieve faster speedup at the prefilling. In particular, to process 1M tokens, CoLM-Air yields nearly $1.6\times$ and $3.0\times$ faster prefilling when $\mathcal{C}$ is $\{16, 16\}$ and $\{8, 8, 8, 8\}$. And, when combining with MInference [28], CoLM-Air achieves at most $27\times$ speedup ($1920 \rightarrow 70.0$) in Figure 5d. These results all demonstrate that our CoLM-Air can effectively speedup prefilling.

## 4.5 Chain Tuning

Benefiting from the causality of CoM architecture, CoLM is composed of multiple chains, that each chain can leverage capability from previous chains. Therefore, based on this property, we propose *Chain Tuning*, which aims to fine-tune the latter chains while freezing the first few chains. Chain

Tuning can partially reduce the tuning costs and mitigate catastrophic forgetting via preserving the first few chains. Moreover, when we adopt the CoLM-Air setting and freeze the first chain, the keys and values from the fine-tuned models can seamlessly transfer to the original model, without any additional computations. Here, we utilize the expanded models (with two chains) trained in Section 4.2 as the baseline, and then fine-tune the expanded chain.

In addition, compared with Parameter-Efficient Fine-Tuning (PEFT) methods (e.g., LoRA [29]), our Chain Tuning is fully compatible with these PEFT methods, while further enable KV sharing to switch between the fine-tuned models and the original model. In other words, we can apply LoRA to the latter chains for even greater parameter efficiency. To demonstrate this capability, we choose GLUE benchmark [30] for a faster validation. We report the results within Table 4, and more experimental details can refer to Appendix A.1. From Table 4, we observe Chain-tuning can boost model performance by just fine-tuning approximately 42% of the model parameters.

Table 4: GLUE dev set results. Each task is reported by accuracy. "+ CT" means Chain Tuning, and the values in brackets mean the fine-tuned parameters and the total parameters.

| Model | SST-2 | COLA | MNLI | MRPC | QNLI | QQP | RTE | WNLI | Avg. |
|---|---|---|---|---|---|---|---|---|---|
| Baseline | 67.89 | 32.98 | 35.87 | 49.02 | 51.77 | 51.65 | 54.87 | 43.66 | 48.46 |
| **+ LoRA** | 91.74 | 58.46 | 81.32 | 61.27 | 73.20 | 54.92 | 55.96 | 52.11 | 66.12 |
| **+ CT** | 92.32 | 54.87 | 82.26 | 62.01 | 82.87 | 55.22 | 59.21 | 53.52 | **67.79** |

## 4.6 Generative Performance

To assess performance on open-ended generation, we instruction-tuned our expanded TinyLLaMA-v1.1 model (from Section 4.2) on the Alpaca dataset [31] and evaluated it on MT-Bench [32]. The results in Table 5 show that our model, which leverages KV sharing, outperforms the similarly tuned baseline, particularly in reasoning and humanities tasks. This validates that the CoM framework and KV sharing mechanism are effective for enhancing instruction-following and conversational abilities.

Table 5: MT-Bench scores for instruction-tuned models. "Ours" refers to our expanded model. Both models are fine-tuned on the Alpaca dataset. Higher scores are better.

| Model | Writing | Roleplay | Reasoning | Math | Coding | Extraction | STEM | Humanities | Avg. |
|---|---|---|---|---|---|---|---|---|---|
| TinyLLaMA-v1.1 | 2.85 | 3.20 | 2.05 | 1.10 | 1.30 | 1.40 | 2.35 | 3.20 | 2.18 |
| Ours | 2.75 | **3.40** | **2.70** | 1.00 | 1.10 | 1.30 | 2.35 | **3.65** | **2.28** |

## 5 Related Works

**Foundation Models** play a critical component in machine learning for effectively modeling relationships among data. From the perspective of architectural level, we posit that existing Large Foundation Models (LFM), can be conceptually decomposed into two parts, Base Architecture and Scaling Architecture. For Base Architecture, it refers to the backbone network (e.g., Transformer [7], Mamba [33], RWKV [34]) to perform feature transformation from the original data. For Scaling Architecture, it represents the scaling strategies (e.g., Increasing layers, dimensions or experts [35, 14]) to improve model capabilities. To some extent, CoM can also be regarded as a scaling architecture via increasing the number of chains, so that it can be applied to any base architectures. Moreover, compared with Dense or MoE architectures, the core idea of CoM is to integrate capabilities across varying scales and allow us to leverage inherited knowledge from previous models to improve scaling efficiency and avoid catastrophic forgetting.

**Elastic Inference** is to offer multiple sub-models tailored to handle diverse user instructions under different deployment constraints. Current LLM architectures can only provide a fixed inference capacity, and thus requires us to pre-train multiple models at different sizes, leading to inefficiencies. To this end, Once-for-All [10] build a super-net and then randomly sample a specialized sub-network for training. Some other works [11, 36] introduce some nested architecture designs into network blocks to offer elastic inference across different model granularity, but also require sampling strategy

of each sub-network. By incorporating causal dependencies among sub-models within CoR, CoM can integrate multi-scale training within a single forward pass, demonstrating better efficiency.

**Continual Training** is to continue the training process from the pre-trained models to boost its capability and avoid catastrophic forgetting. To leverage the capability of existing pre-trained models, the most simple method is using depth-up scaling method (i.e., increase the number of layers), such as Solar 10.7B [37]. Depth-up scaling can effectively reuse weights from previous layers for initialization, but fails to retain the output logits (i.e., probability) of the classification layers. CoM can be regarded as a width-up scaling method, which can perfectly preserves the knowledge of the entire model from the first layer to the final layer. Please note that CoM is also compatible with the depth-up scaling approach. Besides, Progressive neural network (PNN) [12] is a seminal work in continual training, which also motivates us to develop CoM. PNN extends the MLP layers by incorporating new parameters via lateral connections, and then apply to the reinforcement learning tasks. Compared to PNN, CoM extends its scope to a broader range and effectively demonstrates the feasibility of using Transformers for pre-training.

# 6 Conclusion

In this paper, we introduce the concept of "*Chain-of-Representation*" (CoR) to encode multi-scale information within hidden representation. Subsequently, we further propose "*Chain-of-Model*" (CoM) learning to capture the causal relationship across varying scales among CoR features of each layer within the model. The model built upon the CoM framework can provide multi-scale functionality and retain capability from previous scales for usage. Therefore, we devise "*Chain-of-Language-Model*" (CoLM) by integrating the Chain-of-Model (CoM) paradigm into each layer of the Transformer architecture. As a result, CoLM can offer multi-scale sub-models at varying scales within a unified language model. Moreover, we propose a KV sharing mechanism that computes all keys and values within the first chain. This design significantly unleashes the flexibility and extensibility of language models, including prefilling acceleration, dynamic LLM switching and so on. Experimental results across multiple datasets demonstrate that our CoLM family matches the performance of standard Transformers, while offering superior extensibility and adaptability across a wide range of scenarios, paving a new way to build next-generation foundation models.

# 7 Limitations

While Chain-of-Model (CoM) introduces a flexible and efficient paradigm for scaling language models, some challenges and areas for future exploration remain. First, the Chain-of-Linear layer, in its current form, presents complexities for Tensor Parallelism (TP), a key technique for training extremely large models. Although compatible with other parallelism strategies, optimizing CoM for TP-centric infrastructures requires further investigation. Second, in the CoLM-Air architecture, the first chain's quality is critical, as it generates the keys and values for all subsequent chains. Determining the optimal configuration and training strategy for this foundational chain is crucial for maximizing the overall model's potential. We provide a more detailed discussion of these limitations and outline potential future directions in Appendix A.5.

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

# A Appendix

## A.1 Experimental Details

**Datasets**    SlimPajama [22] is a deduplicated, multi-corpora, open-source dataset for training LLMs. We use SlimPajama dataset exclude books corpus as the pre-training corpus, result in approximately 600 billion tokens. The composition of our used dataset is shown in Table 6.

| Dataset | Ratio |
|---------|-------|
| Commoncrawl | 52.2% |
| C4 | 26.7% |
| GitHub | 5.2% |
| ArXiv | 4.6% |
| Wikpedia | 3.8% |
| StackExchange | 3.3% |

Table 6: Dataset statistical of SlimPajama dataset.

**Hyperparameters**    Table 7 presents the hyperparameters that were used in pretraining. Besides, FlashAttention-2 [38] is applied to speed up attention computation. For the Expansion experiment, we set the training step to 20,000, while preserving the same configuration as the pre-training. Table 8 presents the hyperparameters used for Chain Tuning.

**Evaluation**    In this project, we utilized the Eleuther AI Language Model Evaluation Harness library [25]. We used the default evaluation configuration. For chain tuning, we set the prompt template to the default from the evaluation scripts.

Table 7: Hyperparameters used in pretraining.

| Hyperparameter | Value |
|----------------|-------|
| Learning Rate | $2e - 4$ |
| Sequence Length | 4096 |
| Batch Size Per Device | 8 |
| Gradient Accumulation Steps | 8 |
| Optimizer | Adamw |
| LR Scheduler Type | Linear |
| Warmup Steps | 2000 |
| Max Norm | 1.0 |
| Activation checkpointing | full |
| Mixed Precision for Parameters | bfloat16 |
| Mixed Precision for Reductions | float32 |

Table 8: Hyperparameters used in Chain Tuning.

| Hyperparameter | Value |
|----------------|-------|
| Learning Rate | $2e - 4$ |
| Epochs | 1 |
| Batch Size Per Device | 4 |
| Gradient Accumulation Steps | 32 |
| Optimizer | Adamw |
| LR Scheduler Type | Cosine |
| Warmup Ratio | 0.1 |
| Sequence Length | 4096 |

**Model Configuration**    Table 9 presents the detailed configurations for all the models used in our experiments. Each model is identified by its name and characterized by configurations such as the Chain setting, dimension (dim), hidden dimension (hidden dim), the number of layers, attention

Table 9: Model Configuration.

| Model Name | Chains | # Params | Dim | Hidden Dim | # Layers | # Heads | # KV Heads |
|---|---|---|---|---|---|---|---|
| expansion-llama3.2-1B | {32, 8} | 1.93B | 2560 | 10240 | 16 | 40 | 10 |
| expansion-tinyllama | {32, 8} | 1.44B | 2560 | 7040 | 22 | 40 | 5 |
| Llama | {32} | 1.10B | 2048 | 8192 | 16 | 32 | 8 |
| CoLM_16-16 | {16, 16} | 0.86B | 2048 | 8192 | 16 | 32 | 8 |
| CoLM_8-24 | {8, 24} | 0.92B | 2048 | 8192 | 16 | 32 | 8 |
| CoLM_8-8-16 | {8, 8, 16} | 0.80B | 2048 | 8192 | 16 | 32 | 8 |
| CoLM_8-8-8-8 | {8, 8, 8, 8} | 0.74B | 2048 | 8192 | 16 | 32 | 8 |
| CoLM_16-16-same-params | {16, 16} | 1.11B | 2560 | 8192 | 16 | 32 | 8 |
| CoLM_8-8-8-8-same-params | {8, 8, 8, 8} | 1.18B | 3072 | 8192 | 16 | 32 | 8 |

heads, key-value heads and the number of parameters. Figure 12 illustrates the structure of the Chain of Transformer, designed for multi-scale representation learning.

## A.2 Full Results

### A.2.1 Prefilling

Our prefilling is evaluated on a single NVIDIA A100 GPU with a batch size of 1. Specifically, from 128K to 1M, we perform chunk-wise prefilling (e.g., chunk operation for layer calculation) to avoid the out-of-memory (OOM) issue, and we also evaluate and validate our method when combining with MInference [28]. Here, we evaluate six configurations by using different chain settings ({16, 16}, {8, 8, 8, 8}, {8, 8, 16}, {8, 24}) and increase the number of layers to ensure the parameter sizes are comparable. Table 10 and Table 11 report all results of our CoLM-Air setting (1B and 8B) of a sequence length from 2K to 64K. And Table 12 and Table 13 report all results of our CoLM-Air setting (1B and 8B) of a sequence length from 128K to 1M. From these results, we can find that CoLM can significantly speed up the prefilling stage in calculating long-context instructions.

Moreover, our CoLM is also extremely suitable for PD disaggregation architecture [39–41], as it only requires us to deploy the entire weights of the first chain on the prefilling server to calculate KV caches, which significantly reduces the computation loads of the prefilling server.

Table 10: Prefilling results on LLama-3.2-1B setting (2K - 64K). All results are reported in milliseconds. $D_{x_1}$ represents the dimension of the first chain.

| Method | L | $D(D_{x_1})$ | Length | | | | | |
|---|---|---|---|---|---|---|---|---|
| | | | 2K | 4K | 8K | 16K | 32K | 64K |
| LLama-3.2-1B | 16 | 2048 (2048) | 51.0 | 62.0 | 132.0 | 313.0 | 997.0 | 2,600 |
| *Same configuration* (CoLM-Air) | | | | | | | | |
| {16,16} | 16 | 2048 (1024) | 47.0 | 47.6 | 60.0 | 140.0 | 380.0 | 1,189 |
| {8, 8, 8, 8} | 16 | 2048 (512) | 48.0 | 50.0 | 51.0 | 71.0 | 193.0 | 613.0 |
| *Similar parameters* (CoLM-Air) | | | | | | | | |
| {16,16} | 22 | 2048 (1024) | 66.0 | 69.0 | 78.0 | 183.0 | 503.0 | 1,587 |
| {8, 8, 8, 8} | 26 | 2048 (512) | 76.0 | 76.0 | 78.0 | 105.0 | 293.0 | 929.0 |
| {8,8,16} | 24 | 2048 (512) | 68.0 | 69.0 | 73.0 | 99.0 | 272.0 | 863.0 |
| {8,24} | 21 | 2048 (512) | 62.0 | 60.0 | 65.0 | 87.0 | 243.0 | 768.0 |

### A.2.2 CIFAR-10 MLP Experiments

We compare a baseline fully-connected MLP against several Chain-of-Model MLP (CoM-MLP) variants on CIFAR-10. The baseline model uses a 4-layer architecture with hidden sizes [768, 1024, 512, 256] (an input embedding of dimension 768, followed by three dense layers and a 10-classes output). CoM-MLP models augment each layer with multiple sequential conditional modules specified by the "Chain-setting" vector. Models were trained using the Adam optimizer with an initial

Table 11: Prefilling results on LLama-3.1-8B setting (2K - 64K). All results are reported in seconds. $D_{x_1}$ represents the dimension of the first chain. $D_{x_1}$ represents the dimension of the first chain.

| Method | L | D($D_{x_1}$) | Length | | | | | |
|---|---|---|---|---|---|---|---|---|
| | | | 2K | 4K | 8K | 16K | 32K | 64K |
| LLama-3.1-8B | 32 | 4096 (4096) | 0.19 | 0.38 | 0.79 | 1.72 | 4.18 | 11.55 |
| *Same configuration* (CoLM-Air) | | | | | | | | |
| {16,16} | 32 | 4096 (2048) | 0.09 | 0.14 | 0.28 | 0.64 | 1.64 | 4.89 |
| {8, 8, 8, 8} | 32 | 4096 (1024) | 0.09 | 0.10 | 0.12 | 0.28 | 0.72 | 2.30 |
| *Same parameters* (CoLM-Air) | | | | | | | | |
| {16,16} | 43 | 4096 (2048) | 0.13 | 0.18 | 0.37 | 0.85 | 2.17 | 6.49 |
| {8, 8, 8, 8} | 52 | 4096 (1024) | 0.13 | 0.16 | 0.19 | 0.43 | 1.06 | 3.61 |
| {8,8,16} | 48 | 4096 (1024) | 0.13 | 0.15 | 0.17 | 0.34 | 0.94 | 3.35 |
| {8,24} | 40 | 4096 (1024) | 0.11 | 0.14 | 0.15 | 0.85 | 2.17 | 2.82 |

Table 12: Prefilling results on LLama-3.2-1B setting (128K - 1M). All results are reported in seconds. The value in the bracket is using the MInference trick. $D_{x_1}$ represents the dimension of the first chain.

| Method | L | D($D_{x_1}$) | Length | | | |
|---|---|---|---|---|---|---|
| | | | 128K | 256K | 512K | 1M |
| LLama-3.2-1B | 16 | 2048 (2048) | 8.6 (5.5) | 32.4 (10.3) | 132.0 (21.4) | 522.0 (45.8) |
| *Same configuration* (CoLM-Air) | | | | | | |
| {16, 16} | 16 | 2048 (1024) | 4.0 (2.1) | 15.4 (4.5) | 63.2 (9.7) | 265.0 (20.1) |
| {8, 8, 8, 8} | 16 | 2048 (512) | 1.9 (1.0) | 7.5 (2.2) | 30.6 (4.8) | 124.0 ( 9.9) |
| *Same parameters* (CoLM-Air) | | | | | | |
| {16, 16} | 22 | 2048 (1024) | 5.4 (2.9) | 21.0 (6.2) | 88.8 (13.1) | 352.0 (28.4) |
| {8, 8, 8, 8} | 26 | 2048 (512) | 3.2 (1.6) | 12.5 (3.5) | 50.7 ( 7.4) | 213.0 (15.5) |
| {8, 8, 16} | 24 | 2048 (512) | 2.9 (1.5) | 11.3 (3.1) | 46.6 ( 6.7) | 188.0 (14.5) |
| {8, 24} | 21 | 2048 (512) | 2.6 (1.3) | 9.9 (2.8) | 40.7 ( 6.0) | 165.0 (13.3) |

learning rate of $10^{-3}$, a batch size of 64, for a total of 20 epochs. Reported accuracies are the mean $\pm$ standard deviation over 3 independent runs.

Table 14 summarizes the test accuracy and parameter count for the baseline MLP and CoM-MLP configurations. While all models achieve similar accuracy (around 54–55%), the CoM-MLP with Chain-setting $\{4, 12\}$ achieves the best accuracy ($55.11 \pm 0.21\%$) with fewer parameters compared to the dense baseline.

## A.3 Training Loss Curves

We conducted several small-scale experiments (batch size 8, 15K steps) to study how different CoLM configurations and settings affect training convergence. The goal was to understand the impact of CoM design choices on convergence speed and final loss. Figures 6 plot the training loss over time for each experiment.

**Comparison of CoLM Configurations** Figure 6(a) shows the training loss curves for four chain configurations: equal four-chain $\{8, 8, 8, 8\}$, two 16-unit chains $\{16, 16\}$, and two mixed settings $\{8, 24\}$ and $\{8, 8, 16\}$. The 16-16 configuration converges more quickly and reaches the lowest loss, indicating more efficient training, which appears to yield faster convergence and lower loss under these conditions, suggesting a trade-off in chain design.

Table 13: Prefilling results on LLama-3.1-8B setting (128K - 1M). All results are reported in seconds. The value in the bracket is using the MInference trick. $D_{x_1}$ represents the dimension of the first chain.

| Method | L | $D(D_{x_1})$ | Length | | | |
|---|---|---|---|---|---|---|
| | | | 128K | 256K | 512K | 1M |
| LLama-3.1-8B | 32 | 4096 (4096) | 36.0 (21.6) | 134.0 (49.5) | 512.0 (106.0) | 1920 (186.0) |
| *Same configuration* (CoLM-Air) | | | | | | |
| {16, 16} | 32 | 4096 (2048) | 15.6 (6.4) | 60.0 (13.3) | 244.8 (28.5) | 927.0 (89.0) |
| {8, 8, 8, 8} | 32 | 4096 (1024) | 7.3 (2.5) | 28.1 ( 5.1) | 117.3 (10.9) | 472.0 (44.7) |
| *Same parameters* (CoLM-Air) | | | | | | |
| {16, 16} | 43 | 4096 (2048) | 20.6 (10.8) | 77.7 (23.5) | 314.8 (45.6) | 1293 (128.0) |
| {8, 8, 8, 8} | 52 | 4096 (1024) | 12.0 ( 6.2) | 46.4 (12.5) | 192.4 (25.4) | 738 ( 70.1) |
| {8, 8, 16} | 48 | 4096 (1024) | 11.0 ( 5.9) | 40.5 (10.7) | 167.4 (23.9) | 680 ( 65.0) |
| {8, 24} | 40 | 4096 (1024) | 9.2 ( 4.2) | 35.4 ( 9.7) | 146.0 (19.6) | 591 ( 56.1) |

Table 14: CIFAR-10 classification accuracy (mean ± std) and parameter count for the baseline MLP and CoM-MLP variants.

| Model | Chains | #Params | Accuracy (%) |
|---|---|---|---|
| MLP (Dense) | - | 3,806,218 | 54.93 ± 0.23 |
| CoM-MLP | {4, 4, 4, 4} | 3,263,754 | 54.60 ± 0.16 |
| CoM-MLP | {8, 8} | 3,443,978 | 54.68 ± 0.04 |
| CoM-MLP | {4, 12} | 3,534,090 | **55.11 ± 0.21** |
| CoM-MLP | {4, 4, 8} | 3,353,866 | 54.55 ± 0.36 |

**Loss Across Different Heads**  We also examined head-wise training by optimizing each chain output head together in the {8, 8, 8, 8} CoLM setting. Figure 6(c) plots the training loss for each of the four heads. The results reveal that larger chains effectively leverage the outputs of preceding chains, resulting in greater capacity and improved performance. This trend demonstrates that deeper chains benefit from the cumulative computations of earlier chains, enhancing their ability to model complex features. The intermediate heads show intermediate performance, reflecting a gradual increase in capacity along the chain depth. These findings highlight the hierarchical structure of CoLM, where later chains achieve stronger capacity by building on the representations provided by earlier ones.

**Effect of KV Sharing**  Figure 6(c) compares the {8, 8, 8, 8} setting with and without the KV-sharing mechanism (CoLM-Plus). While KV sharing results in a slightly higher loss for the same head, it

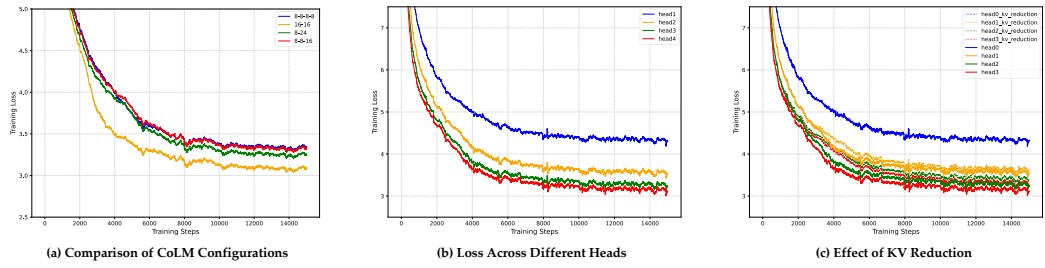

(a) Comparison of CoLM Configurations       (b) Loss Across Different Heads       (c) Effect of KV Reduction

Figure 6: (a) Training loss for different CoLM chain configurations. (b) Training loss for each output head in the {8, 8, 8, 8} model (heads 1–4). (c) Training loss for the {8, 8, 8, 8} configuration, with versus without KV sharing.

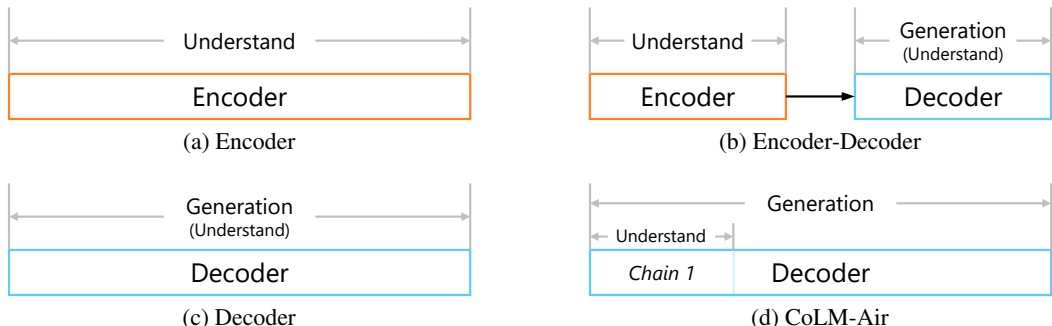

Figure 7: Illustrative comparisons between Encoder, Encoder-Decoder, Decoder and CoLM-Air, in term of understanding and generation capabilities.

significantly improves prefilling speed, as discussed in Section 4.4. This trade-off highlights the efficiency gains of KV sharing without a substantial impact on performance.

## A.4  Discussion

***Question* 1** *Chain-of-model architecture seems to be a major improvement in the width of language models (i.e., dimension). Can it be applied to the depth of language models (i.e., layers)?*

Yes, the existing mechanism of chain-of-model design requires us to first determine the depth of large foundation models, and then allows us to extend its capability from the perspective of the dimension (i.e., width). Actually, we also conduct preliminary experiments to explore the feasibility of applying the idea of chain-of-model across the depth (i.e., layers) of the language model. Here, when applying CoM into the model depth, we also require it to meet two characteristics to ensure the alignment with the settings of our current version (i.e., CoLM):

1. The model should encompass multiple sub-models to enable dynamic selection and inference;
2. Each sub-model can seamlessly reuse past states (e.g., keys and values in Transformer) that are generated by different sub-models.

To meet these requirements, we have made two major designs based on the Transformer architecture: (a) we select some intermediate layers and then append the classifier to each as the sub-models; (b) we limit keys and values are only produced by the first sub-model (the first few bottom layers), which are shared across the subsequent layers. In our internal experiments, the setting (a) can guarantee that the model can be optimized and reach convergence, while we can also obtain multiple nested sub-models. Previous researches (e.g., GoogleNet [42]) also validate the availability of such a solution. But this design also introduces additional complexity, as we need to maintain multiple classifiers, which is extremely cumbersome and redundant. Although we can tie the weights of each classifier together to reduce model parameters, it will also cause interference to the model performance and cannot save any computations to calculate each classification head. However, when further integrated with the setting (b), which delivers the keys and values from the bottom layers to the top layers, we find that it severely degrades model performance. In other words, only using bottom-level states (i.e., keys and values) for all Transformer layers is not enough to create high-level semantics for next-token prediction. Existing attempts [43, 44] also indicate that we can only restrict key and value sharing to adjacent layers, which identifies the importance of depth to abstract high-level feature states (i.e., keys and values). Besides, some recent works (e.g., YOCO [45]) have explored reusing features from the outputs of half of the layers, which introduces modifications that combine efficient self-attention and cross-attention mechanisms. However, this technique is restricted to KV cache reduction during prefilling and does not enable multiple sub-model inference. Some other works (e.g., MoD [46]) introduce routers to dynamically skip certain layers for faster inference. But this design requires us to reset the keys and values to zero for the skipped layers, which potentially introduces discrepancies between training and inference. In our CoM method, we can retain all keys and values at each layer, enabling inference at multiple scales. Generally speaking, to abstract high-level features in neural network, depth usually matters more than width and and preserving invariant depth (i.e., layers) for

foundation models is extremely necessary. This can also explain why our current version can only be applied to width, rather than depth. We will continue exploring solutions that extend the idea of chain-of-model to both width and depth in the future.

***Question 2*** *What are the differences between the Chain-of-Model (CoM) and Mixture-of-Experts (MoE) architectures?*

Table 15: Comparisons between MoE and CoM from different aspects.

|  | **Mixture-of-Expert (MoE)** | **Chain-of-Model (CoM)** |
|---|---|---|
| Expert Capability | Equivalent | Weak $\rightarrow$ Strong |
| Expert Range | FFN | Full Model |
| Expert Activation | Sparse | Nested |

The concept of Mixture of Experts (MoE) was first proposed by [14], which aims to create multiple expert networks and then use a gating mechanism for selective activation. As Transformer has exhibited huge potential in scalability, some works [35, 47] have investigated how to apply the MoE architecture into the FFN layers within Transformer to build the giant network. On the basis of this, some works [6, 48] have successfully scaled up language models with MoE architectures to a substantial size. We deem that MoE can also be considered as a kind of scaling architecture. Existing MoE architectures in Transformer are designed to create multiple experts with similar capabilities at the FFN layers, while only a few experts are activated for each token prediction. In the CoM structure, it aims to create a sequence of nested experts, progressing from weak to strong capability. Specifically, each expert contains the capability of all its preceding experts (i.e., weaker experts). In Table 15, we also illustrate the comparisons between CoM and MoE architectures. Therefore, CoM and MoE are designed from different perspectives to construct experts. In other words, CoM is completely orthogonal to MoE, meaning both of them can be applied within the same architecture and inherit advantages from each. We will explore how to combine CoM and MoE architecture, and leave it as future work.

***Question 3*** *When do we need the Chain-of-Model architecture?*

Generally, Chain-of-Model (CoM) architecture can be considered as an innovative design to optimize the scaling architecture (e.g., increasing model capabilities) of foundation models, rather than altering the backbone network (e.g., Transformer). Therefore, it can be generalized to any model with different architectures, such as CNNs or parallelized RNNs. However, this does not mean we can apply CoM architecture to any scenario. Based on the designs of CoM architecture, it can be considered as a composition of multiple nested sub-models. Each sub-model will contribute to the capability of the subsequent models. More specifically, the first chain (sub-model) plays a critical role in CoM architectures as it directly determines the understanding capability of the entire model. Therefore, if the capacity of the model is not sufficient to exhibit the power of the scaling law or lacks enough generalization, it is unnecessary to use the architecture of the chain-of-model method. In other words, the CoM structure is better suitable for large-scale model sizes, rather than smaller ones. This also indicates that CoM is an advanced design to optimize the scaling architecture.

## A.5   Limitations

From the perspective of methodology, Chain-of-Model (CoM) introduces an innovative and brand-new solution to revolutionize the scaling architecture of large-scale foundation models. However, CoM still remains some ongoing challenges when applying it to practical applications:

- Although the CoM architecture offers exceptional flexibility in utilizing LLMs, it also poses new challenges for training a large-scale CoM architecture at the infrastructure level. Specifically, Chain-of-Linear layer is compatible with data, pipeline, and context parallelism (DP, PP, CP), but not well-suited for tensor parallelism (TP). In the naive implementation (Please see Figure 2b or Algorithm 1), the Chain-of-Linear layer can be viewed as a composition of multiple sub-linear layers. As a result, if we apply tensor parallelism to each sub-linear within the Chain-of-Linear

layer, it will significantly increase the number of all-reduce operations and data access when compared with the standard Linear. Although block-wise sparse kernel optimization can help us to alleviate these issues, we still need to devise a new pipeline mechanism when handling large tensor inputs. To some degree, Chain-of-Linear layer can be considered as an imbalanced split-wise tensor, which requires us to specifically design a tensor parallelism mechanism for it. Here, we leave it as future work.

- Empowered by chain-of-model design, CoLM is a novel architecture that combines multiple nested language models at varying scales. Benefiting from such a design, each model will inherit all its nested sub-models and extend their capability. In other words, each nested model will affect its subsequent model. More specifically, CoLM further introduces a KV-reduction mechanism to guarantee that the understanding parts (e.g., keys and values in Transformer) are all produced by the first chain. That means the first chain plays a very important role in building CoLM as it determines the understanding capability of the entire model. So, compared to the other chains, we expect the first chain to pay more attention to expressive and generalized content states (i.e., keys and values), in relative to its generation. Therefore, how to explore the optimal setting for the first chain could be very important in the CoLM framework. However, due to resource limitations, we leave it as future works.

---

**Algorithm 1** Pseudo Code for Chain-of-Linear Layer (Linear_chain.py)

---

```python
class Linear(nn.Module):

    def __init__(
        self, dim: int, hidden_dim: int, chains: List[int], bias = False
    ):
        """
            dim:        Dimension of input features
            hidden_dim: Dimension of output features
            chains:     Base of each chain
            bias:       whether using bias or not
        """
        super().__init__()
        self.num_chain = len(chains)
        self.in_dims = [dim * c // sum(chains) for c in np.cumsum(chains)]
        self.out_dims = [hidden_dim * c // sum(chains) for c in chains]
        self.mlps = nn.ModuleList([
            nn.Linear(indim, outdim, bias=bias)
            for indim, outdim in zip(self.in_dims, self.out_dims)
        ])

    def forward(self, x: torch.Tensor, chain: int = None) -> torch.Tensor:
        """
            x:          Input features, shape = (B, L, D)
            chain:      The number of activated chains ([0, num_chain - 1])
        """
        if chain is None:
            chain = self.num_chain - 1

        assert (
            0 <= chain < self.num_chain
        ), f"The chain id should be in [0, {self.num_chain})."
        outputs = [
            self.mlps[i](x[..., :indim])
            for i, indim in enumerate(self.in_dims[:chain + 1])
        ]
        return torch.cat(outputs, dim=-1)
```

---

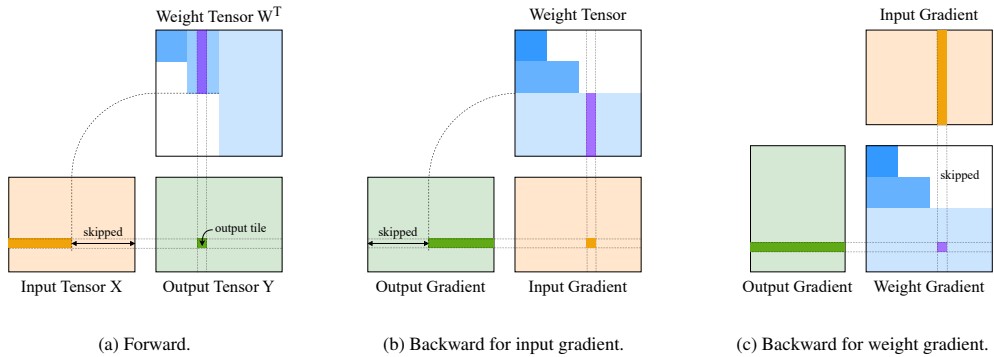

|  |  |  |
|:---:|:---:|:---:|
| (a) Forward. | (b) Backward for input gradient. | (c) Backward for weight gradient. |

Figure 8: Block-sparse GeMM kernel for Linear (Chain) layer.

Table 16: Speed comparisons between different kernels. The "fwd", "$\mathrm{bwd_w}$" and "$\mathrm{bwd_i}$" represent the speed of forward, backward (weight) and backward (inputs). Chain-of-Linear (Naive) refers to Algorithm 1, and Chain-of-Linear (Triton) corresponds to our block-wise sparse kernel implementation.

| | | | Computation | | |
|---|---|---|---|---|---|
| Implementation | Dim | Params | fwd | $\mathrm{bwd_w}$ | $\mathrm{bwd_i}$ |
| $\mathcal{C} = \{8, 8, 8, 8\}$, *tensor size is (4096, 4096).* | | | | | |
| Linear | 4096 | 167K | 0.67 | 0.62 | 0.64 |
| Chain-of-Linear (Naive) | 4096 | 104K | 0.58 | 0.49 | 0.72 |
| Chain-of-Linear (Triton) | 4096 | 104K | 0.47 | 0.46 | 0.56 |
| $\mathcal{C} = \{4, 4, 4, 4, 4, 4, 4, 4\}$, *tensor size is (8192, 8192).* | | | | | |
| Linear | 8192 | 671K | 1.56 | 1.61 | 1.56 |
| Chain-of-Linear (Naive) | 8192 | 419K | 0.96 | 0.84 | 1.82 |
| Chain-of-Linear (Triton) | 8192 | 419K | 0.77 | 0.76 | 0.90 |

## A.6 Implementation

In this section, we will provide detail implementation about each module used in CoLM, including Linear, Attention, FFN, normalization and so on.

### A.6.1 Linear

In Algorithm 1, we provide the naive PyTorch implementation of Chain-of-Linear layer. Generally, the proposed Chain-of-Linear layer can be viewed as a composition of multiple sub-linear layers. Under the same dimension, our implementation (i.e., Algorithm 1) can reduce more weights when compared with the standard Linear layer, but it will add the number of data loading and all-reduce communication. Therefore, we further devise a block-wise sparse kernel to accelerate the computation of Chain-of-Linear structure, which is introduced in below.

**Block-wise Sparse Kernel Acceleration**   We developed a block-wise sparse matrix multiplication kernel for both forward and backward paths of Chain-of-Linear layer. To conserve GPU memory, we store the weights of Linear (Chain) layer in block-compressed-sparse-row (BCSR) format and perform matrix multiplication only on the step-shaped mask to accelerate computation. The kernel implementation is based on the Triton General-Matrix-Multiplication (GeMM) kernel and supports both sparse input and sparse output. For sparse input, each thread block processes a row in the BCSR matrix, skipping unmapped data from the other dense input matrix. For sparse output, thread blocks in the masked area will exit immediately. Figure 8 illustrates the details of our block-wise sparse kernel for acceleration. We also report the computation comparisons between our kernel with naive

**Algorithm 2** Pseudo Code for Attention (Chain) Layer.

```python
from flash_attn import flash_attn_func
from Linear_chain import Linear

class Attention(nn.Module):

    def __init__(
        self, dim: int, n_head: int, n_kv_head: int, chains: List[int]
    ):
        super().__init__()
        assert sum(chains) == n_head
        for chain in chains:
            assert chain * n_kv_head % n_head == 0
        self.head_dim = dim // n_head

        self.wq = Linear(dim, dim, chains)
        self.wk = Linear(dim, n_kv_head * self.head_dims, chains)
        self.wv = Linear(dim, n_kv_head * self.head_dims, chains)
        self.wo = Linear(dim, dim, chains)

    def forward(self,
                x: torch.Tensor,
                freqs_cis: torch.Tensor,
                chain: int = None) -> torch.Tensor:
        bsz, seqlen, dim = x.shape

        q, k, v = self.wq(x, chain), self.wk(x, chain), self.wv(x, chain)
        q = q.view(bsz, seqlen, -1, self.head_dim)
        k = k.view(bsz, seqlen, -1, self.head_dim)
        v = v.view(bsz, seqlen, -1, self.head_dim)

        q, k = apply_rotary_emb(q, k, freqs_cis)

        o = flash_attn_func(q, k, v, causal=True)
        o = o.view(bsz, seqlen, -1)
        return self.wo(o, chain=chain)
```

implementation in Table 16. From Table 16, we observe that the naive implementation is exceedingly slow in computing backpropagation gradients for the input data, while our kernel significantly reduces its computation for input data, and then improves the speed of both the forward and backward stages. Besides, our kernel can also reduce the number of all-reduce communications during the distributed training. Benefiting from this design it allows us to conduct large-scale pre-training efficiently.

### A.6.2 Attention

Just as aforementioned in Section 3.2.1, we can replace all linear layer (i.e., Q, K, V, O) by using Chain-of-Linear layer. For MHA setting [7], we use $\mathcal{C} = \{c_1, \ldots, c_n\}$, where $\sum_i^n c_i = h$, to determine how many heads (i.e., $q_i$, $k_i$, $v_i$) should be assigned to each chain. For the GQA setting [20], we require $c_i \times h_{kv}$ should be a multiple of $\text{sum}(\mathcal{C})$. For Chain-of-Attention with KV sharing mechanism, we require each $c_i$ should be a multiple of $h_{kv}$, as all keys and values are computed within the first chain and then shared across the remaining chains. And then, we will replicate keys and values per chain to guarantee that their number to match the number of query. Figure 9 illustrates the differences between GQA and Attention (Chain) with the KV sharing mechanism. We also provide the PyTorch implementation of Chain-of-Attention in Algorithm 2 and its version with KV sharing in Algorithm 3.

**Algorithm 3** Pseudo Code for Attention (Chain) Layer with KV sharing.

```python
import torch.nn as nn

from flash_attn import flash_attn_func
from Linear_chain import Linear

class Attention(nn.Module):

    def __init__(
        self, dim: int, n_head: int, n_kv_head: int, chains: List[int]
    ):
        super().__init__()
        assert sum(chains) == n_head
        for chain in chains:
            assert chain % n_kv_head == 0
        self.head_dim = dim // n_head
        self.kv_dim = chains[0] * self.head_dim

        self.wq = Linear(dim, dim, chains)
        self.wk = nn.Linear(self.kv_dim, n_kv_head * self.head_dims, chains)
        self.wv = nn.Linear(self.kv_dim, n_kv_head * self.head_dims, chains)
        self.wo = Linear(dim, dim, chains)

    def forward(self,
                x: torch.Tensor,
                freqs_cis: torch.Tensor,
                chain: int = None) -> torch.Tensor:
        bsz, seqlen, dim = x.shape
        x0 = x[..., :self.kv_dim]

        q, k, v = self.wq(x, chain), self.wk(x0), self.wv(x0)
        q = q.view(bsz, seqlen, -1, self.head_dim)
        k = k.view(bsz, seqlen, -1, self.head_dim)
        v = v.view(bsz, seqlen, -1, self.head_dim)

        q, k = apply_rotary_emb(q, k, freqs_cis)

        n_rep = q.shape[2] // k.shape[2]
        k = k.repeat(1, 1, n_rep, 1)
        v = v.repeat(1, 1, n_rep, 1)
        o = flash_attn_func(q, k, v, causal=True)
        o = o.view(bsz, seqlen, -1)
        return self.wo(o, chain=chain)
```

### A.6.3 FFN

In Figure 10, we illustrate the differences between the original FFN and our proposed FFN layer, which just needs to replace the Linear layer by using a Chain-of-Linear layer. They are equivalent when the number of chains $n = 1$. We use the same hyperparameter $\mathcal{C}$ of the Attention module to set up the FFN layer, so that the output features of the Transformer block (i.e., Attention + FFN) also follow the criteria of CoL in Definition 2. We also offer a PyTorch implementation of FFN (GeLU) in Algorithm 4 and FFN (SwiGLU) in Algorithm 5 for reference.

**Algorithm 4** Pseudo Code for FFN implementation (GeLU).

```python
import torch.nn.functional as F
# Import Chain-of-Linear Layer
from Linear_chain import Linear

class FFN(nn.Module):

    def __init__(
        self, dim: int, hidden_dim: int, chains: List[int]
    ):
        """
            dim:        Dimension of input features
            hidden_dim: Dimension of output features
            chains:     Base of each chain
        """
        super().__init__()

        self.w1 = Linear(dim, hidden_dim, chains=chains)
        self.w2 = Linear(hidden_dim, dim, chains=chains)

    def forward(self, x: torch.Tensor, chain: int = None) -> torch.Tensor:
        return self.w2(F.gelu(self.w1(x, chain)), chain)
```

**Algorithm 5** Pseudo Code for FFN implementation (SwiGLU).

```python
import torch.nn.functional as F
# Import Chain-of-Linear Layer
from Linear_chain import Linear

class FFN(nn.Module):

    def __init__(
        self, dim: int, hidden_dim: int, chains: List[int]
    ):
        """
            dim:        Dimension of input features
            hidden_dim: Dimension of output features
            chains:     Base of each chain
        """
        super().__init__()

        self.w1 = Linear(dim, hidden_dim, chains)
        self.w2 = Linear(hidden_dim, dim, chains)
        self.w3 = Linear(dim, hidden_dim, chains)

    def forward(self, x: torch.Tensor, chain: int = None) -> torch.Tensor:
        return self.w2(F.silu(self.w1(x, chain)) * self.w3(x, chain), chain)
```

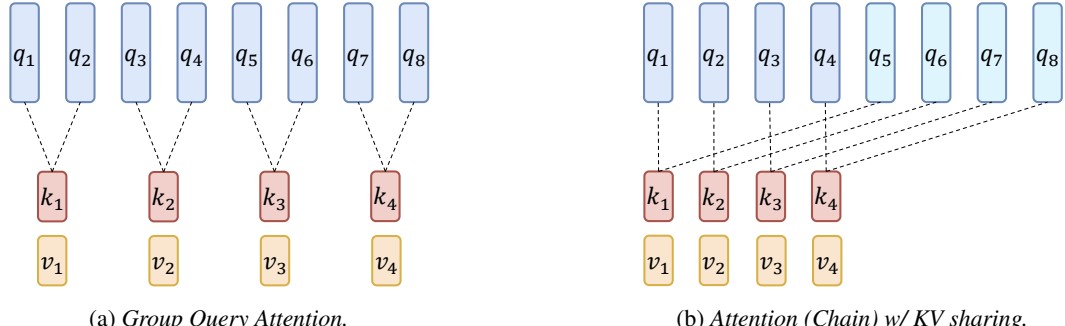

(a) *Group Query Attention.*    (b) *Attention (Chain) w/ KV sharing.*

Figure 9: Differences between Group Query Attention (GQA) and Attention (Chain) with KV sharing in the attention implementation.

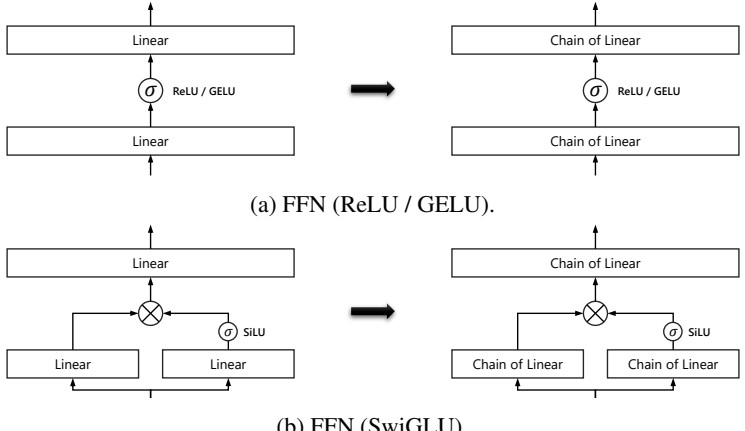

(a) FFN (ReLU / GELU).

(b) FFN (SwiGLU).

Figure 10: FFN for CoR. We replace all Linear by using Chain-of-Linear layer.

### A.6.4 Embedding

Figure 11 illustrates how the Embedding layer is utilized in our CoLM architecture by activating different numbers of chains. To use different scales, we only need to activate the dimension corresponding to its all preceding chains.

**Algorithm 6** Pseudo Code for Embeddings.

```python
import torch.nn as nn

class Embedding(nn.Embedding):

    def __init__(
        self,
        dim: int,
        vocab_size: int,
        chains: List[int],
        padding_idx: Optional[int] = None,
    ):
        super().__init__(vocab_size, dim, padding_idx)
        self.dims = [dim * c // sum(chains) for c in np.cumsum(chains)]
        self.num_chains = len(chains)

    def forward(self, x: torch.Tensor, chain: int = None):
        if head is None:
            return F.embedding(x, self.weight, self.padding_idx)
        else:
            assert chain < self.num_chains
            dim = self.dims[chain]
            return F.embedding(x, self.weight[:, :dim], self.padding_idx)
```

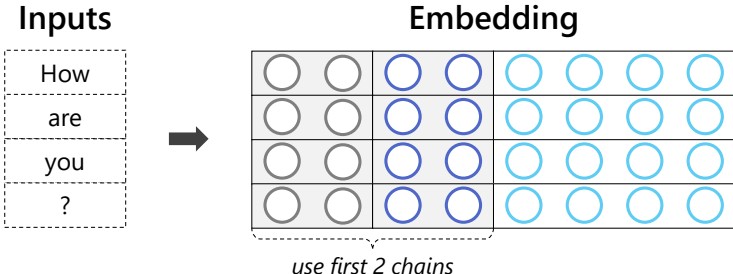

*use first 2 chains*

Figure 11: Example of Embedding in CoLM architecture. In this example, we design the number of chains as 3, while the dimensions for each chain are set as $\{2, 2, 4\}$. If we expect to use the first 2 chains, we only need to activate the first 4 neurons for each word in the embedding layer.

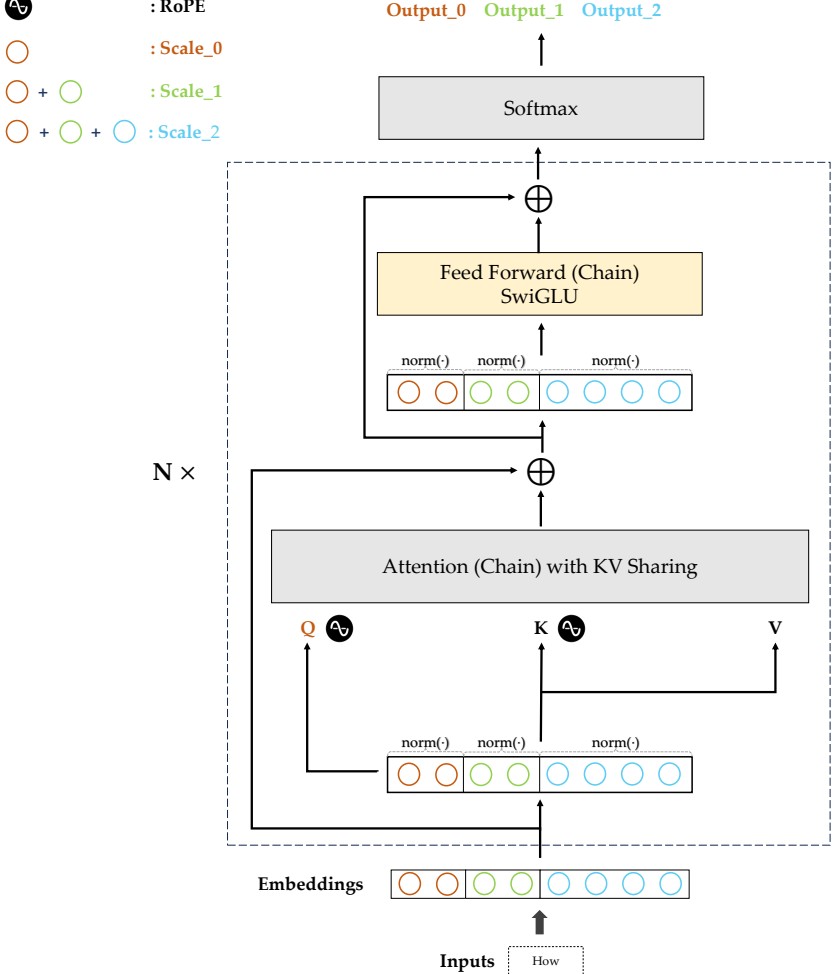

Figure 12: The architecture of our CoLM. The model use Chain-of-Representation to build each layer, including embedding, self-attention, feed-forward network, normalization. Each scale contributes progressively refined outputs (e.g., Output0, Output1, Output2).

