# OpenReview forum: "Chain-of-Model Learning for Language Model"
_NeurIPS.cc/2025/Conference — NeurIPS 2025 poster_

### Official Review · Reviewer_Uwqq · 2025-06-27

**Clarity:** 3
**Significance:** 2
**Originality:** 3
**Rating:** 4
**Confidence:** 3

**Summary:**

This paper proposes **Chain-of-Model (CoM)**, a progressive learning paradigm inspired by Progressive Neural Networks, where newly added neurons are causally dependent on the preceding ones. CoM is applied to progressively expand the **width** of Transformer models. The authors provide detailed implementations for each module of the Transformer architecture (e.g., FFN, multi-head attention, normalization, and embedding). Experimental results demonstrate the effectiveness of CoM in model expansion, elastic inference, and continual learning.

**Questions:**

See Weaknesses

**Ethical Concerns:**

["NO or VERY MINOR ethics concerns only"]

**Final Justification:**

The authors' response addresses my concerns. I recommend accepting this paper.

**Limitations:**

See Weaknesses

**Quality:**

3

**Strengths And Weaknesses:**

**Strengths**

- The proposed method is intuitive and well-motivated for scaling the width of Transformers. The design is supported by experimental evidence.
- The paper provides a clear and detailed description of the method, including pseudocode and implementation details.
- The experiments cover a variety of scenarios, such as model expansion, tuning, and inference.
- The paper is well-written and easy to follow.


**Weaknesses**

While the method is novel and well-implemented, its **superiority over existing approaches remains unclear**. Although the experiments cover multiple settings, **comparisons with strong baselines are missing**, making it difficult to assess the practical advantages of CoM.

1. **Comparison in 'Chain Expansion' (Section 4.2)**:
    The authors should compare CoM with other model expansion techniques, such as vanilla width expansion.

   > For example, when expanding the representation space from NNN to N+MN+MN+M, CoM adds a matrix of shape (N+M,M)(N+M, M)(N+M,M). How would this compare to simply adding a matrix of shape (M,M)(M, M)(M,M) without causal constraints?
   >  This ablation could help validate the core contribution of CoM—namely, the necessity of causal dependencies across chains.

2. **Comparison in 'Chain Tuning' (Section 4.5)**:
    The paper should include comparisons with parameter-efficient fine-tuning methods such as **LoRA**, to show how CoM performs in terms of efficiency and performance in limited-resource settings.

3. **Lack of comparison with MoE**:
    The authors claim that CoM is compatible with Mixture-of-Experts (MoE) models. However, a natural question arises:

   > Is it more effective to use CoM to increase the capacity of each expert, or to increase the number of experts directly?
   >  Both approaches aim to scale model capacity—some comparison or analysis would be valuable.

4. **Concerns about evaluation tasks**:
    All evaluated benchmarks appear to be **multiple-choice tasks**. It would be beneficial to include results on **generative tasks**, which may better reflect the model's performance in real-world use cases.




**Overall**

This work addresses an important problem: how to progressively scale Transformer models without retraining from scratch. The method is well-structured and the engineering implementation is solid. However, to fully validate its contribution, the paper should include **comparisons with relevant baseline methods** and consider **broader evaluation settings**, including generative tasks.

---

> ### Author Rebuttal · Authors · 2025-07-31
>
> Thank you for your valuable comments. Below are our response to address your concerns:
>
> 1. **Comparison CoM with other model expansion techniques, such as vanilla width expansion. (W1)**:
>
> - Thank you for this insightful question. Generally, vanilla width expansion (i.e., CoM without constraints) can be considered as creating an individual expert (model) and combining it into the final prediction layer. Thus, this expert cannot leverage the capability from the previous model, which could limits its upper bound. To resolve your concerns, we also offer some experiments to compare with vanilla width expansion (i.e., CoLM without constraints). The results are shown in below:
>
>     | Model                   | HellaSwag | Obqa | WinoGranda | ARC-e| ARC-c | Boolq | Piqa | Sciq | Avg   |
>     |------------------------|------------|-------|-------------|--------|--------|--------|-------|-------|--------|
>     | Baseline               | 61.47      | 36.62 | 59.43       | 55.47  | 32.68  | 55.99  | 73.56 | 84.20 | 57.43  |
>     | + Vanilla Width Expansion | 61.20      | 35.41 | 60.22       | 55.33  | 32.08  | 57.61  | 73.72 | 85.00 | 57.57  |
>     | + Expansion (Ours)            | 61.66      | 36.62 | 60.62       | 56.27  | 32.94  | 58.44  | 74.05 | 86.20 | **58.35** |
>
> - Besides, to some degree, vanilla width expansion (i.e., without constraints) can also be considered as a specical case of Chain-of-Model, that $y_i$ is only conditioned on $x_i$ which belong to a subset of Definition 2. In our CoLM, our normalization also remove this causal constraint to slightly improve training efficiency. Therefore, we deem that for some specific module, we can use vanilla width expansion (CoM without constrains) to add some efficiency. In the future, we will also continue to seek more GPU resource to explore the optimal setting for our CoLM architecture.
>
> 2. **Comparisons with parameter-efficient fine-tuning methods such as LoRA, to show how CoM performs in terms of efficiency and performance in limited-resource settings. (W2)**:
>
> - Thank you for this excellent suggestion to clarify the relationship between Chain Tuning and established PEFT methods as LoRA.
>     Chain Tuning is not merely another PEFT method, but rather a novel fine-tuning paradigm that leverages the inherent causal structure of our Chain-of-Model (CoM) architecture. As described in lines 290-292, our Chain Tuning strategy involves freezing the initial chain(s) and only fine-tuning the subsequent ones. Therefore, its perfectly preserve the previous module to generate keys and values. Consequently, in KV sharing, the KV cache generated by the original pre-trained model can be perfectly and seamlessly reused by any model that has been fine-tuned on the latter chains. To the best of our knowledge, this is the first work that allows us to seamlessly switch different LLMs without re-computing KV caches. In contrast, the motivation of LoRA is designed to reduce the heavy fine-tuning cost over the huge model, which is also compatible with our method, as we can apply LoRA to our latter chains only. The below Table also summarizes the comparisons between our Chain Tuning and LoRA.
>     | Model            | SST-2 | COLA  | MNLI  | MRPC  | QNLI  | QQP   | RTE   | WNLI  | Avg.  |
>     |------------------|-------|-------|-------|-------|-------|-------|-------|-------|--------|
>     | Baseline         | 67.89 | 32.98 | 35.87 | 49.02 | 51.77 | 51.65 | 54.87 | 43.66 | 48.46  |
>     | LoRA             | 91.74 | 58.46 | 81.32 | 61.27 | 73.20 | 54.92 | 55.96 | 52.11 | 66.12  |
>     | + CT             | 92.32 | 54.87 | 82.26 | 62.01 | 82.87 | 55.22 | 59.21 | 53.52 | **67.79** |
>
>     In the future, we will also add these discussion into our final version to make a clear understanding and highlight our biggest advantages of chain tuning.
>
> 3. **Comparison with MoE. (W3)**:
>
> - Thanks for your question. We want to highlight that CoM and MoE are two strategies for model scaling from different dimensions, just as discussed in Question 2 of A.4 in the Appendix. CoM focuses on progressive scaling and offers multi-scale abilities. MoE is to increase model capability via building giant model with sparse activation among one scale. That means, MoE is good at enhancing model capability within single scale, while CoM is to add new scale over existing scales. For example, if we want to extend an additional chain (scale) over existing models, we can still use MoE within this chain (scale), but bridge the connections among multiple scales via CoM. We will add more discussions about this part to further highlight the differences between CoM and MoE.
>
> 4. **Concerns about generation tasks. (W4)**:
>
> - We are grateful to the reviewer for this insightful suggestion. To address this point, we have conducted new experiments focusing on generative instruction-following. Here, we took the TinyLlama-v1.1 model as our baseline and use our chain expansion model for comparison. Both the baseline and our expanded model were then instruction fine-tuned on the Alpaca dataset.
>     We evaluated the generative performance of both models on MT-Bench [1], a standard benchmark for assessing conversational and instruction-following abilities. The results are reported in below:
>     | Model            | Writing | Roleplay  | Reasoning  | Math  | Coding  | Extraction |STEM | Humanities  | Avg.  |
>     |------------------|-------|-------|-------|-------|-------|-------|-------|-------|--------|
>     | Tinyllama         | 2.85 | 3.20 | 2.05 | 1.10 | 1.30 | 1.40 | 2.35 | 3.20 | 2.18  |
>     | Ours             | 2.75 | 3.40 | 2.70 | 1.00 | 1.10 | 1.30 | 2.35 | 3.65 | **2.28** |
>
>     These results also indicate that our method can also achieve better performance on generation tasks than baseline, which validates the generalization of our proposed method. We will add these results into our final version to further highlight the effectivenss of our method on generation tasks.
>     > [1] Judging LLM-as-a-Judge with MT-Bench and Chatbot Arena, NeurIPS 2025

---

> > ### Comment · Area_Chair_ZaV6 · 2025-08-05
> > **Discussion**
> >
> > Dear Reviewer Uwqq,
> >
> > The authors have responded to your concerns. How does their response change your view of the paper? If it does not, please clarify what the authors can do to address your concerns. If it does, please consider adjusting your score based on their response.
> >
> > Your AC

---

> > ### Comment · Reviewer_Uwqq · 2025-08-06
> > **Thanks for your rebuttal**
> >
> > Dear authors,
> >
> > I appreciate your detailed rebuttal. It has sufficiently addressed my concerns, and I have adjusted my score to reflect this.

---

> > > ### Author Response · Authors · 2025-08-06
> > >
> > > Dear Reviewer Uwqq,
> > >
> > > We would like to express our sincere gratitude for your detailed and thoughtful review of our manuscript. Your in-depth feedback and expert evaluation were instrumental in helping us strengthen the rigor and clarity of our work.
> > >
> > > We have made every effort to address all of your comments thoroughly. If any part of our response remains unclear or requires further elaboration, we would be more than willing to provide additional clarification or revisions.
> > >
> > > Thank you again for your valuable input and for contributing to the improvement of our work.

---

### Official Review · Reviewer_xjCS · 2025-06-30

**Clarity:** 3
**Significance:** 4
**Originality:** 3
**Rating:** 5
**Confidence:** 3

**Summary:**

This paper proposes a novel learning paradigm, termed "Chain-of-Model" (CoM), which aims to address the core challenges in the model scaling and deployment of current Large Language Models (LLMs). Its core idea originates from "Chain-of-Representation" (CoR), which treats the representation vectors in the model's hidden layers as a concatenation of multiple "sub-chains". The authors  apply this paradigm to the Transformer architecture, designing the "Chain-of-Language-Model" (CoLM). Experimental results show that this method performs slightly worse than the vanilla Transformer in terms of effectiveness, but has advantages in incremental learning, elastic inference, and efficient deployment.

**Questions:**

See Weaknesses

**Ethical Concerns:**

["NO or VERY MINOR ethics concerns only"]

**Final Justification:**

I think this paper should be accepted to Neurips 2025. Most of my concerns regarding the comprehensiveness of experiments and the practical application of this method have been addressed through the authors' detail responses.

**Quality:**

3

**Strengths And Weaknesses:**

**Strengths:**

- The "Chain-of-Representation" idea proposed in this paper is highly innovative. The resulting model is inherently capable of multi-scale adaptation, which can solve many problems in incremental learning, elastic inference, and efficient deployment.

- The paper provides a thorough discussion on various aspects of the Chain-of-Representation concept, including model training  inference, etc.

- The experimental results demonstrate that the proposed method achieves good performance.

**Weaknesses:**

- Potential Impact of KV Sharing on Generation Quality: The paper mentions that the KV sharing mechanism "slightly affects the performance" , a claim verified by the accuracy on commonsense reasoning tasks. However, the specific impact of this on open-ended text generation tasks (e.g., in terms of coherence, logic, and creativity) remains unclear. Since higher-order chains rely on the semantic information (K and V) generated by lower-order chains, this could lead to an information bottleneck. Providing some qualitative analysis or results on generation-focused benchmarks would allow for a better assessment of the trade-offs of this design.

- Compatibility Issues with Tensor Parallelism (TP): The authors acknowledge in the appendix that the naive implementation of the Chain-of-Linear layer is not well-suited for Tensor Parallelism (TP), which is a very practical obstacle for training very large-scale models. Although the authors mention optimization via a custom-developed kernel, the paper does not provide a clear answer as to whether the additional communication overhead brought by TP can be fully resolved. This could be a significant barrier to the widespread adoption of this architecture in the industry.

---

> ### Author Rebuttal · Authors · 2025-07-31
>
> Thank you for your valuable comments. Below are our response to address your concerns:
>
> 1. **Potential Impact of KV Sharing on Generation Quality. (W1)**:
>
> - We sincerely thank you for providing these constructive comments for us. We also admit that adding generation results can better demonstrate the effectiveness of our method. To resolve your concerns, we also provide our results on generation tasks. Specifically, we took the TinyLlama-v1.1 model as our baseline and applied our expanded model (used in Section 4.2), which use KV sharing to reuse KV cache from the TinyLlama. Both baseline and our model are fine-tuned on Alpaca datasets for post-training. We report the generation results on MT-Bench[1], a standard benchmark for assessing conversational and instruction-following abilities. The results are shown below:
>
>     | Model            | Writing | Roleplay  | Reasoning  | Math  | Coding  | Extraction |STEM | Humanities  | Avg.  |
>     |------------------|-------|-------|-------|-------|-------|-------|-------|-------|--------|
>     | Tinyllama         | 2.85 | 3.20 | 2.05 | 1.10 | 1.30 | 1.40 | 2.35 | 3.20 | 2.18  |
>     | Ours             | 2.75 | 3.40 | 2.70 | 1.00 | 1.10 | 1.30 | 2.35 | 3.65 | **2.28** |
>
>     These results also indicate that by using KV sharing, our model can significant enhance performance over the original model. We will add these generation results into our final version to make a clear understanding.
>
> - Besides, just as mentioned in Line 200 ~ 204, KV sharing mechanism used in KV sharing can be considered as a trade-off technique built based on the standard CoLM architecture, to balance performance and additional flexibility / efficiency. For the standard CoLM architecture, our method can achieve comparable performance to the baseline model. Since KV sharing mechanism can significantly reduce the prefilling speed (nearly 27x) and enable seamlessly LLM switching across different scales, we can increase the training budget of CoLM-Air to further improve its capability. In the future, we will continue to search the optimal setting for the first chain to design our CoLM-Air. Overall, we will highlight this detail in our final version to make a clear understanding.
>
> 2. **Compatibility Issues with Tensor Parallelism. (W2)**:
>
> - Thanks for your comments. Generally, our method is fully compatible with existing parallelisms, including Data Parallelism (DP), Pipeline Parallelism (PP), Context Parallelism (CP) and so on. In our experiments, we also fully utilized Sharded Data Parallel (FSDP) to pre-train our model. For Tensor Parallelism (TP), our method can also use it to each chain actually by using Algorithm 1 in Appendix A.6.1. It will not introduce any additional computation cost, but increases some communication cost in data access during the distributed training. Therefore, just as discussed in Appendix A.5, we propose a block-wise sparse kernel based on triton to mitigate this problem. In the future, we will also continue to devise novel parallelism methods (e.g., chain parallelism) to enable more efficient scalabilty for our method.

---

> > ### Comment · Reviewer_xjCS · 2025-08-06
> > **Thanks for your reply**
> >
> > Dear authors,
> > Thanks for your detailed reply. Most of my concerns have been well addressed.

---

> > > ### Author Response · Authors · 2025-08-06
> > >
> > > We sincerely appreciate the thoughtful and detailed review you provided for our manuscript. Your critical insights and careful evaluation played a vital role in refining and improving our work.
> > >
> > > We have made every effort to address your comments comprehensively, and we hope our responses meet your expectations. If there are any remaining concerns or if further clarification is needed, we would be grateful for the opportunity to continue the dialogue and make additional improvements.
> > >
> > > Thank you once again for your valuable contribution.

---

> ### Comment · Area_Chair_ZaV6 · 2025-08-05
> **Discussion**
>
> Dear Reviewer xjCS,
>
> The authors have responded to your concerns. How does their response change your view of the paper? If it does not, please clarify what the authors can do to address your concerns. If it does, please consider adjusting your score based on their response.
>
> Your AC

---

### Official Review · Reviewer_LJrW · 2025-07-03

**Clarity:** 3
**Significance:** 3
**Originality:** 3
**Rating:** 4
**Confidence:** 4

**Summary:**

In summary, the authors propose a novel learning paradigm called "Chain-of-Model" (CoM). Within this paradigm, the hidden representations are divided into multiple sub-representations, each of which represents a different scale. With such division of hidden representations, the authors propose the concept of "Chain-of-Layer" (CoL), where the scales are learned causally within a layer. That is, the scale $i$ of the layer output only depends on the scales $\leq i$ of the layer input. By stacking layers that satisfy the CoL property, the entire model becomes CoM. Based on this concept, the authors redesign the common layers for language models to satisfy the CoL property and propose Chain-of-Language-Model (CoLM). The proposed CoLM enjoys several properties: 1) The existing LLMs can be treated as chain=1 and easily extended by adding another chain without training from scratch. 2) A model trained with CoLM can be deployed with dynamic inference capacity. 3) CoLM models with KV sharing achieve better pre-filling latency. These properties are evaluated on 1B scale model pertaining with several common-sense reasoning tasks.

**Questions:**

1. The proposed CoLM is mainly tested on Llama variants. Is the proposed CoLM applicable to other model series such as Qwen and DeepSeek MLA?

2. For the chain expansion experiment, each model is only expanded with 1 chain. It would be good to see the performance of extending more chains (e.g., $c_2=4, c_3=4$) and discuss the pros/cons.

**Ethical Concerns:**

["NO or VERY MINOR ethics concerns only"]

**Final Justification:**

In general, this paper proposes a novel learning scheme to scale up pre-trained models. During the rebuttal, most of my concerns were resolved. The remaining concern is mainly about the scalability of the proposed method to larger models. So I will maintain my recommendation as borderline accept.

**Limitations:**

Yes

**Quality:**

3

**Strengths And Weaknesses:**

Strength:
1. This paper studys an important problem as scaling LLM models is extremely resource-consuming. The proposed CoLM could extend an existing model with more chains (more capacity) without training from scratch, which provides a valuable solution for the community
2. Though there are several weaknesses, the proposed method is well-presented in the paper and looks quite novel to me

Weakness:
1. The proposed CoLM is mainly validated on models with 1B parameters, which seems not enough to guarantee that it will remain the same performance for large scales such as 3B, 8B or even larger. Though it is understandable that pre-training 3B/8B model is resource-restricted, it would be nice to see experiments where the proposed CoLM is used to extend some larger models such as Llama-3.2-3B and Llama-3.1-8B.
2. In Table 2, although the CoLM expansion introduces additional 0.8B parameters, the improvement of LLaMA-3.2-1B looks trival to me given that it is trained with 8B tokens.
3. Since the proposed CoLM makes changes to the layer structure. It is unclear whether the proposed method is compatible to the existing large-scale training techniques (e.g., Different parallelism).
4. As the author mentioned, each scale should have its own loss function and that potentially limits the training efficiency when the number of chains is huge
5. With the KV sharing mechanism, the key/value heads only contain the information of the first scale. When the model is expanded with multiple rounds, the model capacity will be limited by the key/value heads.

---

> ### Author Rebuttal · Authors · 2025-07-31
>
> Thank you for your valuable comments. Below are our response to address your concerns:
>
> 1. **It is better to see experiments on 3B, 8B or larger (W1)**:
>
> - Thanks for your constructive comments. We also agree that the experiments on 3B, 8B will be better to demonstrate the potential of CoLM-Air. Besides, since the first chain of CoLM is still the standard language model, it naturally maintains the scalability of existing language models. To validate this point, we conduct experiments using CoLM (chain=[8, 8, 8, 8]) configurated with 4096 dims and 32 layers as LLama-7B configuration, resulting in a model with 4.1B parameters (Chain-of-Layer requires fewer parameters that dense layer). Besides, to make a fair comparison, we also set a dense model (4096 Dims and 16 Layers), with nearly 4.1B parameters. Due to resource limitations (32x 40GB A100 GPUs), the batch size is set as 2 per GPUs for 4.1B model and 1 per GPUs (accum_step=2) for 7B models, seq length is 2048. We apply FSDP and the total training step is 15K steps during the rebuttal period, and we report PPL over SlimPajama datasets per 5K steps to verify its convergence. The results are reported in below:
>
>     | Model |Bsz| Memory | Dim | Layer | Params | 5K | 10K | 15K |
>     |-|-|-|-|-|-|-|-|-|
>     | CoLM (8-8-8-8) | 2|26GB | 4096 | 32 | 4.1B | 22.56 | 17.01  | 15.89 |
>     | CoLM-Air (8-8-8-8) |2| 26GB | 4096 | 32 | 4.1B | 22.74 | 17.45 | 16.48 |
>     | Dense |2| 39GB | 4096 | 20 | 4.1B | 23.29 | 17.12 | 15.80 |
>     | Dense |1| 32GB | 4096 | 32 | 7.3B | 22.26 | 16.39 | Ongoing |
>
>   Due to resource limitations, the total pre-training is still under-fit, and we will continue to seek more GPU resources in the future to fully pre-train a larger size LLM over 7B for usage. Moreover, as shown in the above table, the CoLM configurations require less GPU memory. Therefore, within the same computation, our COLM allows more training steps. We also provide codes in the supplemental materials to reproduce our experiments. We will also put these experiments into our paper to highlight the scalability of our model.
>
> 2. **The improvement of LLaMA-3.2-1B looks trival (W2)**:
>
> - Thanks for your comment and for closely examining our experimental results. The main reason for the marginal improvement over LLama-3.2-1B is because we only use SlimPajama (670B tokens) as the pre-trained dataset and only pre-train 20K steps for pre-training with total 8B tokens. In the original LLama-3.2-1B, it uses high-quality dataset with over 15T tokens plus massive pre-training. Therefore, it will be a little difficult to signficantly improve LLama performance with total 8B tokens training, but it still achieves nearly 0.15 points improvement. To this end, we also offer results on TinyLLama-v1.1, which is built upon SlimPajama plus Starcoder dataset, to make a fair comparison. In this case, we observe that our model can improve the baseline of TinyLlama-v1.1 with nearly 0.92 points within 8B tokens training. This results also indicate our method can effectively increase model scales in a progressive way. In the future, we will also continue to increase our pre-training budget to further increase the capability of our model.
>
> 3. **Whether the proposed method is compatible to the existing large-scale training techniques. (e.g., Different parallelism) (W3)**:
>
> - Thanks for your comments. Generally, our method is fully compatible with existing parallelisms, including data, pipeline, context and expert parallelism (i.e., DP, PP, CP and EP). In our experiments, we also use Fully Sharded Data Parallel (FSDP) to pre-train our model. For Tensor Parallelism (TP), our method actually is also compatible with it by using TP over each chain based on Algorithm 1. It will not introduce any extra computation but increase some communication cost (e.g., more all-reduce to access data). Therefore, we use triton to implement a block-wise sparse kernel to improve our training efficiency in Appendix A.5. We will also explore more efficient parallelism methods (e.g., chain parallelism) to scale up our model in the future.
>
> 4. **Each scale should have its own loss function and that potentially limits the training efficiency when the number of chains is huge. (W4)**:
>
> - Thank you for this astute observation. We admit this point and thereby adopt the MRL [1] trick to increase the training efficiency for multiple chains (Line 213-215). MRL trick is to freeze the backbone network (pre-trained) and then only fine-tune the classification layer to provide multi-scale prediction. Motivated by MRL [1], we also first pre-train the whole network, and then freeze the backbone and only fine-tune classification layer with multiple chains to increase training efficiency. We will provide more details about this part to make a clear understanding in the final version.
>
>     > [1] matryoshka representation learning. NeurIPS 2022
>
> 5. **When the model is expanded with multiple rounds, the model capacity will be limited by the key/value heads. (W5)**:
>
> - Thank you for this insightful question. We admit your concern, just as our discussion in the Limitation part (Linear 650-661), that the first chain plays a critical role in CoLM-Air setting and how to determine the optimal setting for the first chain is important. Specifically, KV sharing can be considered as a trade-off technique (Line 200-204) built based on the standard CoLM architecture, to balance performance and extra flexibility / efficiency. For the standard CoLM without KV sharing, our method can achieve comparable performance to the baseline. Specifically, as CoLM takes fewer GPU memory / flops and KV sharing mechanism can significantly reduce the prefilling speed (27x) with seamlessly LLM switching, we can increase the training budget of CoLM-Air to further improve its capability. In the future, we will also continue to explore the optimal setting of the first chain in our CoLM-Air.
>
> 6. **Q1: Is the proposed CoLM applicable to other model series such as Qwen and DeepSeek MLA?**:
>
> - Yes, the idea of CoM framework can be applied to any model series including Qwen and DeepSeek MLA. Based on the Corollary 2.1, any model can be considered as a kind of CoM when n = 1. Therefore, CoM is a general scaling paradigm, which just requires each layer to meet the criteria of chain-of-layer. For Qwen or DeepSeek series model, we just need to replace all Linear layer, attention and normalization by using our setting in Section 3.2. When the number of chain is 1, it is identical to the original LLM settings like Qwen or DeepSeek.
>
>
> 7. **Q2: For the chain expansion experiment, each model is only expanded with 1 chain. It would be good to see the performance of extending more chains and discuss the pros/cons.**:
>
> - Thanks for your constructive comment. Due to resource limitations, in our original experiments (Sec 4.2), we only extend one chain over 1B model to verify the effectiveness of our method in chain expansion, as it introduces more parameters (e.g., 0.8B parameters) for computations. We agree that adding more chains would better highlight its pros and cons. Therefore, we provide multi-chain experiments with 4 chains (C = [8, 8, 8, 8]) on our pre-training settings (Section 4.1). We also provide the detailed training curve about different chains in Figure 6(b) in Appendx, to offer the visualization of using more chains. These results also indicate that using more chains allows model to reduce its training loss. Generally, the pros / cons can be summarized as below:
>
>     - **Pros**: When we use multiple chains, it enables us to conduct elastic inference and efficient progressive scaling to offer different model sizes to meet the requirements from diverse devices. Moreover, by introducing KV sharing mechanism, it can significantly reduce the prefilling speed. Besides, each new chain would build upon the cumulative knowledge of all preceding chains (including the original model), leading to a more powerful model.
>
>     - **Cons**: Just as mentioned in Line 651 - 661, each added chain will reuse knowledge from previous chains. Therefore, if the former chains are not well-trained, they will also affect the latter chains to make prediction. Therefore, how to ensure high-quality train over each chain is still very imporant.
>
>     Overall, multi-chain expansion can be considered as a flexible strategy to extend model and offer elastic inference. We will also add these discussions into our final version to make a clear understanding.

---

> > ### Comment · Area_Chair_ZaV6 · 2025-08-05
> > **Discussion**
> >
> > Dear Reviewer LJrW,
> >
> > The authors have responded to your concerns. How does their response change your view of the paper? If it does not, please clarify what the authors can do to address your concerns. If it does, please consider adjusting your score based on their response.
> >
> > Your AC

---

> > ### Comment · Reviewer_LJrW · 2025-08-05
> > **Thanks for the rebuttal**
> >
> > I want to thank the authors for their detailed responses! Most of my concerns have been addressed. Regarding the scalability to larger models and more chains, I am still concerned but I appreciate the additional experiments and the detailed discussion provided by the authors, which I find reasonable. I completely understand that scaling requires time and resources. Therefore, I will maintain my positive score for this paper.

---

> > > ### Author Response · Authors · 2025-08-06
> > >
> > > We sincerely appreciate the effort and time you have invested in meticulously reviewing our manuscript and offering insightful feedback. Your expertise has significantly contributed to enhancing the quality of our work.
> > >
> > > We are writing to ensure that we have successfully addressed all your concerns to your satisfaction. Should there be any remaining queries or any aspects of our rebuttal that require further clarification, we are fully prepared to engage in further discussion or implement additional revisions as necessary.
> > >
> > > Thank you once again for your suggestions for our work. We look forward to any further insights you may wish to share.
> > >
> > > Best regards,

---

### Official Review · Reviewer_fgKQ · 2025-07-03

**Clarity:** 3
**Significance:** 3
**Originality:** 3
**Rating:** 4
**Confidence:** 4

**Summary:**

The paper proposes Chain-of-Model (CoM) learning: split every hidden vector into ordered “chains,” force each output chain i to depend only on the first i input chains (Chain-of-Layer, CoL) and stack such layers to obtain a network that contains several nested sub-models. The instantiated Chain-of-Language-Model (CoLM) keeps the first chain identical to a standard Transformer and adds extra chains that can be trained later (chain expansion) or dropped at inference time (elastic inference) without recomputing KV caches thanks to a KV-sharing trick (CoLM-Air) .

**Questions:**

1- Which exact normalization scheme do you use (RMSNorm vs. LayerNorm)? Please include the details of how did you adapt the layer norm in your method.

2- Does CoLM (with and without KV sharing) maintain accuracy in long sequence scenarios?

3- How does the method behave at ≥7 B parameters? Do the small accuracy gap of CoLM-Air persist?

**Ethical Concerns:**

["NO or VERY MINOR ethics concerns only"]

**Limitations:**

The paper has no separate limitation section. Including a short “Limitations & Societal Impact” subsection covering the points such as Scalability, Long-context quality etc. would make the submission more transparent and align with NeurIPS best practices.

**Quality:**

3

**Strengths And Weaknesses:**

Strength:

1- The paper is well-written and it is easy to understand. The application is technically sound and potentially useful as it shows an architectural innovation with tangible speed.

2- The method is innovative and tries to tackle an interesting problem.

3- Demonstrates that a single checkpoint can serve multiple width targets with negligible quality loss.

Weaknesses:

1- Some low level details are missing. For example in section 3.2.3 the details of normalisation scheme has not explained.

2- Performance does decline as more chains are added, and even the two-chain CoLM-Air lags the dense baseline by ~0.3 points. Although that margin is still modest (well under one point), it indicates that the KV-sharing trick may exact a small but measurable accuracy cost, which could widen at larger model scales or on harder benchmarks. Demonstrating results at ≥7 B parameters (or providing significance tests to show whether 0.3 is within noise) would help establish how serious the penalty really is.

3- Based on the speed-up curves in Figure 5, latency gains emerge only when the pre-fill prompt exceeds roughly 4 K tokens; below that, CoLM is nearly as slow as the dense baseline. This implies the architecture is chiefly advantageous in long-context settings. Yet all reported generalization scores rely on short-context LM-Harness benchmarks or GLUE tasks. To demonstrate practical value, the authors should also measure long-context accuracy (e.g., on Needle-In-a-Haystack or LongBench) to verify that KV-sharing retains quality.

---

> ### Author Rebuttal · Authors · 2025-07-31
>
> Thank you for your valuable comments. Below are our response to address your concerns:
>
> 1. **Missing the details of normalization. (W1 & Q1)**:
>
> - Thank you for pointing out this missing details. Generally, for each chain-of-representation feature, we apply normalization at the chain-wise level to maintain its Chain-of-Layer property. That means, each chain will have its individual normalization layer, which could be Layer Norm, RMS Norm or others. In our experiment, we use RMSNorm to aligh the Llama setting, which is also widely used in existing LLMs. We apologize for the missing information and will fix it in the final version.
>
> 2. **Demonstration Results over 7B parameters or significant tests on CoLM-Air. (W2 & Q3)**:
>
> - Thanks for your constructive comments. We also agree that the experiments over 7B will be better to demonstrate the potential of CoLM-Air. Besides, since the first chain of CoLM is still the standard language model, it naturally maintains the scalability of existing language models. To validate this point, we conduct experiments using CoLM-Air (chain=[8, 8, 8, 8]) configurated with 4096 dims and 32 layers as LLama-7B configuration, resulting in a model with 4.1B parameters (Chain-of-Layer requires fewer parameters that dense layer). Besides, to make a fair comparison, we also set a dense model (4096 Dims and 16 Layers), with nearly 4.1B parameters. Due to resource limitations (32x 40GB A100 GPUs), the batch size is set as 2 per GPUs for 4.1B model and 1 per GPUs (accum_step=2) for 7B models, seq length is 2048. We apply FSDP and the total training step is 15K steps during the rebuttal period, and we report PPL over SlimPajama datasets per 5K steps to verify its convergence. The results are reported in below:
>
>     | Model |Bsz| Memory | Dim | Layer | Params | 5K | 10K | 15K |
>     |-|-|-|-|-|-|-|-|-|
>     | CoLM (8-8-8-8) | 2|26GB | 4096 | 32 | 4.1B | 22.56 | 17.01  | 15.89 |
>     | CoLM-Air (8-8-8-8) |2| 26GB | 4096 | 32 | 4.1B | 22.74 | 17.45 | 16.48 |
>     | Dense |2| 39GB | 4096 | 20 | 4.1B | 23.29 | 17.12 | 15.80 |
>     | Dense |1| 32GB | 4096 | 32 | 7.3B | 22.26 | 16.39 | Ongoing |
>
>     Due to resource limitations, the total pre-training is still under-fit, and we will continue to seek more GPU resources in the future to fully pre-train a larger size LLM over 7B for usage. Moreover, as shown in the above table, the CoLM configurations require less GPU memory. Therefore, within the same computation, our COLM allows more training steps. But to make a fair comparison, we still compare different settings under the same steps. We also provide codes in the supplemental materials to reproduce our experiments.
>
> - For significant test, we also calculate p-value of CoLM-Air, where p $\approx$ **0.02** < 0.05. These results also indicate the effectiveness of our method. Since our current method is only pre-trained with 200B tokens, we will continue to pre-train our model to see its robustness when using more computational resources. Besides, just as mentioned in Line 200 ~ 204, KV sharing mechanism used in CoLM-Air can be considered as a trade-off technique built based on the standard CoLM architecture, to balance performance and additional flexibility / efficiency. For the standard CoLM architecture, our method can achieve comparable performance to the baseline model. Since CoLM takes fewer GPU memory / flops, and KV sharing mechanism can significantly reduce the prefilling speed (27x) with seamlessly LLM switching, we can increase the training budget of CoLM-Air to further improve its capability. Overall, in the future, we will continue to seek more GPU resources to explore its scalability.
>
> 3. **Prefilling below 4K and Long-context accuracy (W3 & Q2)**:
>
> - For the prefilling, we report the results of CoLM-Air under a batch size of 1 for evaluation (Please see Appendix A.2.1). Therefore, our method cannot fully utilize GPU under 4K with a batch size of 1. However, for sequence length below 4K, our CoLM-Air allows larger batch size when compared with the baseline, as we only need to activate the first chain. Here, our CoLM-Air (8-8-8-8 setting) enables 4x batch size when compared with baseline under 4K settings. We will add these details about prefilling into our final version to make a clear understanding.
> - Besides, in the original experiments, we mainly generate KV caches by using sliding window mask. To extend its long-context ability, we also need to pre-train CoLM-Air on long-context data. Here, we freeze the first chain and then fine-tune its latter chains (i.e., Chain Tuning) on long context, so that it can ensure all keys and values are identical to the original model and thus perserve the original results under short context (only switching to tuned chains when requiring long-context scenarios). We use LongFormer trick [1] to continue pre-train our model on all chains except the first one. And the results on LongBench is shown as:
>
>     | Text Length | Single-Doc QA | Multi-Doc QA| Summarization | Few-shot Learning | Synthetic Task | Code Completion |
>     |-|-|-|-|-|-|-|
>     | 8k | 0.50 | 0.51| 0.52| 0.55 | 0.55 | 0.66 |
>     | 16k | 0.51 | 0.54 | 0.53 | 0.58 | 0.51 | 0.66 |
>     | 32k | 0.53 | 0.58 | 0.56 | 0.61 | 0.53 | 0.67|
>     | 64k | 0.55 | 0.6 | 0.56 | 0.61 | 0.54 | 0.66 |
>
> - These results indicate our method can effectively extend its long-context capability and maintain short-context preformance. In the future, we will investigate better long-context training techniques on CoLM-Air to unlock its potential on long-context tasks.
>     > [1] Longformer: The Long-Document Transformer.
>
> 4.  **Limitations Section**
>
> - We apologize for the misplacement of the Limitation section, which is currently in Appendix A.5. We will move it to the main paper in the final version.

---

> > ### Author Response · Authors · 2025-08-06
> >
> > Dear AC:
> >
> > Thank you very much for your help in engaging the reviewers. We truly appreciate your support throughout the process.
> >
> > Best regards,
> >
> > The authors

---

> > ### Comment · Reviewer_fgKQ · 2025-08-07
> >
> > Thank you for the detailed response.
> > The explanations for concerns 1 and 2 are clear and adequately addressed. However, regarding concern 3, while the authors provide LongBench results for CoLM-Air with chain tuning, the absence of baseline comparisons and limited clarity on the experimental setup remain issues. It is unclear how the long-context experiments were conducted, specifically, whether techniques like YaRN were used to extend the sequence length. As long-context efficiency is a key contribution of the paper, further experiments such as Needle-in-a-Haystack and a more comprehensive evaluation are needed to substantiate the generalization ability of the proposed approach in this setting.

---

> ### Author Response · Authors · 2025-08-07
>
> Dear Reviewer fgKQ
>
> Thank you for your valuable feedback. We are pleased that our previous responses have addressed your first two concerns, and we apologize for the misunderstanding regarding the long-context setting in CoLM-Air. Below are our response to address your final concern:
>
> 1. Generally, the design of current CoLM-Air setting can mainly improve the prefilling speed as it only needs to activate the first chain to calculate keys and values. However, its current architecture is not specific designed for long-context training, which still requires other training tricks (e.g., sparse attention, streaming llm) to enable long-context. In the original paper, our first chain still use the standard language model settings (Line 274), and we report the prefilling speed from 4K to 1M by using sliding window mask tricks (in our response) to demonstrate its efficiency in prefilling speed. Therefore, in the original experiment setup, we mainly validate the efficiency of CoLM-Air in long-context prefilling (via sliding window mask trick), rather that its long-context performance. Just as our previous response, CoLM-Air enables faster prefilling above 4K tokens and supports larger batch size below 4K. We will add these details in the final version to make a clear understanding.
> 2. Besides, in our previous response, we also provide additional long-context experiments and just want to verify that our method can also be combined with existing long-context techniques to seamlessly extend its long-context capability via chain tuning. We also sorry for not reporting the baseline of our previous response, as we want to illustrate that our method can seamless extend its long-context capability via chain tuning by leveraging existing long-context tricks (e.g., sparse attention). Below are our results for comparisons, which reports acc under different length training over CoLM-Air by using sparse attention with chain tuning. Vanilla is the baseline.
>
> | Model | Text Length | Single-Doc QA | Multi-Doc QA| Summarization | Few-shot Learning | Synthetic Task | Code Completion |
> |-|-|-|-|-|-|-|-|
> | CoLM-Air (Vanilla) | 8k | 0.33 | 0.33| 0.34 | 0.47 | 0.46 | 0.50 |
> | |16k | 0.16 | 0.17 | 0.18 | 0.30 | 0.31 | 0.34 |
> | |32k | 0.08 | 0.09 | 0.09 | 0.15 | 0.16 | 0.17|
> | | 64k | 0.04 | 0.04 | 0.05 | 0.05 | 0.04 | 0.06 |
> | CoLM-Air (w/ Sparse Attention) | 8k | 0.50 | 0.51| 0.52| 0.55 | 0.55 | 0.66 |
> | |16k | 0.51 | 0.54 | 0.53 | 0.58 | 0.51 | 0.66 |
> | |32k | 0.53 | 0.58 | 0.56 | 0.61 | 0.53 | 0.67|
> | | 64k | 0.55 | 0.6 | 0.56 | 0.61 | 0.54 | 0.66 |
>
> These results can guarantee our method can improve its long-context capability when combining with existing technologies. Besides, to demonstrate the compatibility of our method with existing long-context techniques, we also presents the combining results with StreamingLLM on PG19 and arXiv.
> |Length | PG19 | arXiv |
> |-|-|-|
> | 8k | 11.08 | 16.77 |
> | 8k + streamingLLM | 10.53 | 16.56 |
> | 16k | 9.59 | 13.76 |
> | 16k + streamingLLM | 9.04 | 13.71 |
> | 32k | 8.48 | 9.62 |
> | 32k + streamingLLM | 7.97 | 9.41 |
> | 64k | 8.02 | 9.15 |
> | 64k + streamingLLM | 7.65 | 8.86 |
>
>
> We want to highlight that to enhance the long-context ability of CoLM-Air, it still requires the integration of existing long-context tricks (e.g., YaRN, Sparse Attention, StreamingLLM and so on). Therefore, we admit that to achieve STOA results, it is still necessary to explore how to combine the advanced long-context methods into CoLM-Air settings, but this task is not the main goal of our paper.
>
>
> Overall, in our paper, we want to claim the main advantages of CoLM-Air is its prefilling efficiency, rather than its long-context capability. Our original experiments mainly use sliding window mask and then evaluate its prefilling speed under long-context data. We apologize for making this misunderstanding. We will add these discussions into our final manuscript to make a clear understanding and avoid any misunderstanding. In the future, we will also consider to conduct more experiments like Needle-In-a-Haystack to explore the potential setting of our CoLM architecture in long context tasks.
>
> Thank you once again for your insightful engagement, which has helped us improve the clarity of our paper.

---

> ### Author Response · Authors · 2025-08-08
>
> Dear Reviewer fgKQ,
>
> We hope our latest response can address your concerns. Besides, to further understand the mechanism of our method in improving prefilling speed, we additional provide the computational complexity of CoLM-Air in sequence calculation.
>
> For a batch (size = B) of sequences with length of L, the complexity of Transformer (Dimension=$D_{x}$) to process this batch is as $(BL^2{D_{x}} + BL{D_{x}}^2)$, where $BL^2{D_{x}}$ represents the complexity of attention and $BL{D_{x}}^2$ represents the complexity of FFN. So, for CoLM-Air, since it only requires the calculation of the first chain to obtain keys and values. Therefore, the Transformer complexity of CoLM-Air for prefilling is as $(BL^2{D_{x_1}} + BL{D_{x_1}}^2)$, where $D_{x_1}$ denotes the dimension of the first chain, just as shown in below Table:
> || Vanilla | CoLM-Air|
> |-|-|-|
> |Attention|$BL^2{D_{x}}$|$BL^2{D_{x_{1}}}$|
> |FFN|$BL{D_{x}}^2$|$BL{D_{x_{1}}}^2$|
>
> As $D_{x_1} << D$, for the prefilling tasks, CoLM-Air enables longer sequence processing within the same batch, or enables larger batch size with processing same sequence length. In other words, CoLM-Air offers a different solution to optimize complexity from the perspectives of Dimension (i.e., $D$), rather than $L$. Therefore, the current architecture of CoLM-Air is not specific designed for long-context setting, and our original experiments mainly use sliding window mask tricks and highlight its advantages in prefilling speed. But just as discussed in Line 274, the first chain is still the standard language model. Hence, you can apply any long-context tricks (e.g., YaRN, Sparse Attention) to extend its capability. Therefore, we conduct some experiments to verify its compatibility with long-context settings, when compared with vanilla model (as it can maintain short-context capability and then improve its long-context capability). As CoLM-Air architecture is not designed for long-context training (i.e., optimize $L$), it still requires us to explore the optimal setting when combining CoLM-Air with existing long-context techniques to achieve STOA results. As CoLM is a new scaling architecture, we will consider this task as the future work to explore its potential and we will add more discussion into our final version to highlight the differences of our method between prefilling and long-context capability.
>
> Thank you once again for your insightful feedbacks.

---

> ### Author Response · Authors · 2025-08-08
>
> Dear Reviewer fgKQ,
>
> Thanks for your comment and we hope our previous response can address your concerns. To further clarify your remaining concern regarding long-context performance, we would like to offer a concrete example to explain the mechanism of our model.
>
> As mentioned in Section 4.4, our CoLM-Air can reduce the prefilling speed as we only need to calculate all keys and values within the first chain and then share it to the latter chains. Therefore, the capability of our model to handle long sequences is dependant on the design of the first chain. In the original setting, our model is trained with 4K, and thus we report the prefilling speed by using sliding window masks. Conversely, if the first chain is a model proficient in long-context tasks, our model can also inherit its advantages.
>
> To make a better understanding, we perform the "Needle in a haystack" on the baseline LLaMA-3.2-1B and our expanded version. We can find their results are completely equivalent, because our expanded version also reuse the keys and values from the first chain and inherit its capability in processing long-sequence:
>
> | Model | Benchmark | Metric | Score |10K|20K|30K|40K|50K|60K|70K|80K|90K|100K|110K|120K|
> | :- | :- | :- | :- |:-|:-|:-|:-|:-|:-|:-|:-|:-|:-|:-|:-|
> | LLaMA-3.2-1B (Baseline) | Needle in Haystack | Exact Match (0-shot) | 96.8 |98|98|96|98|96|98|92|92|90|98|94|96|88|92|
> | **Ours (Chain Expansion)** | Needle in Haystack | Exact Match (0-shot) | **96.8** |**98**|**98**|**96**|**98**|**96**|**98**|**92**|**92**|**90**|**98**|**94**|**96**|**88**|**92**|
>
> From these results, we can find that the long-context capability of our method comes from the first chain. Our first chain can also use some other methods to further improve its long-context capability (e.g., YaRN, StreamingLLM, Sparse Attention and so on). For model itself, our CoLM-Air design mainly reduces the prefilling speed by only using the first chain. Besides, just as mentioned in Line 124, the first chain could be any model, which proves the flexibility and generality of our model.
>
> We hope our experimental explanation can resolve your concerns and we will add these discussions into our final version to make a clear understanding about our design. Thank you for your comments.

---

### Note · Authors · 2025-08-14

Dear Area Chair and Reviewers,

We sincerely thank you for your insightful feedback, which has been invaluable in strengthening our work. We appreciate the recognition of the following strengths:
- **Novelty and Innovation**: Our "Chain-of-Model" paradigm was consistently highlighted as highly innovative, intuitive, and technically sound.
- **Practical Importance**: The potential of our method to address challenges in model scaling, elastic inference, and deployment was highlighted as a valuable contribution.
- **Thorough Methodology**: The detailed description of our method and comprehensive experiments were also appreciated.

In response to the feedback, we have followed their suggestion to further improve our paper during the rebuttal period including:
- **Scalability and Larger-Scale Experiments**: We conducted new experiments on larger models to demonstrate the scalability and memory efficiency of our approach.
- **Generation Quality**: We added evaluations on generative tasks using the MT-Bench benchmark to assess the impact of our method on open-ended text generation.
- **Comparisons to Existing Methods**: We provided experimental comparisons with alternative model expansion techniques and clarified the distinct advantages of our "Chain Tuning" paradigm relative to PEFT methods like LoRA.
- **Clarifications on Methodology**: We added important details regarding our implementation, such as the normalization scheme, compatibility with tensor parallelism, and the objective function for multi-chain training.

We are pleased that these efforts have successfully resolved the concerns of all reviewers and also engaged in a detailed and productive discussion with Reviewer fgKQ. Through extensive clarifications and additional experiments, we were able to resolve what we identified as a misunderstanding regarding our primary claim. We clarified that one of our main contributions is to improve the prefilling efficiency, rather than long-context capability, as CoLM-Air only need to activate the first chain to obtain all keys and values. For long-context tasks, it is equal to the capability of the first chain as it can be any architectures. Hence, you can apply any long-context tricks (e.g., YaRN and Sparse Attention) over it.

We are confident that our Chain-of-Model learning introduces a valuable architectural paradigm for more efficient and scalable LLM development. Thank you for the opportunity to improve our paper through this rigorous review process.

---

### Decision · Program_Chairs · 2025-09-17

**Decision:**

Accept (poster)

**Comment:**

All reviewers agreed this paper should be accepted: it addresses an interesting problem, it is well-written, and the proposed idea creatively addresses the problem. The main concerns were around experimental comparisons: e.g., reviewer Uwqq wanted to see comparisons against other model expansion techniques and parameter-efficient fine-tuning methods. The authors provided these comparisons in the rebuttal. This satisfied the reviewer and they increased their score. In fact, the authors responded to all reviewers extensively, often running detailed experimental comparisons. This addressed additional reviewer concerns: the paper is a clear accept. Authors please remember to include all reviewer requests into the camera-ready paper. Thank you!